# A vaccine targeting mutant IDH1 in newly diagnosed glioma

Michael Platten[1,2,3 ✉], Lukas Bunse[1,2], Antje Wick[4,5], Theresa Bunse[1,2], Lucian Le Cornet[6], Inga Harting[7], Felix Sahm[8,9], Khwab Sanghvi[1], Chin Leng Tan[1], Isabel Poschke[1,3], Edward Green[1], Sune Justesen[10], Geoffrey A. Behrens[11], Michael O. Breckwoldt[7], Angelika Freitag[6], Lisa-Marie Rother[6], Anita Schmitt[12], Oliver Schnell[13], Jörg Hense[14], Martin Misch[15], Dietmar Krex[16], Stefan Stevanovic[17], Ghazaleh Tabatabai[18], Joachim P. Steinbach[19], Martin Bendszus[7], Andreas von Deimling[8,9], Michael Schmitt[12] & Wolfgang Wick[4,5,20]

Mutated isocitrate dehydrogenase 1 (*IDH1*) defines a molecularly distinct subtype of diffuse glioma[1–3]. The most common *IDH1* mutation in gliomas affects codon 132 and encodes IDH1(R132H), which harbours a shared clonal neoepitope that is presented on major histocompatibility complex (MHC) class II[4,5]. An IDH1(R132H)-specific peptide vaccine (IDH1-vac) induces specific therapeutic T helper cell responses that are effective against IDH1(R132H)⁺ tumours in syngeneic MHC-humanized mice[4,6–8]. Here we describe a multicentre, single-arm, open-label, first-in-humans phase I trial that we carried out in 33 patients with newly diagnosed World Health Organization grade 3 and 4 IDH1(R132H)⁺ astrocytomas (Neurooncology Working Group of the German Cancer Society trial 16 (NOA16), ClinicalTrials.gov identifier NCT02454634). The trial met its primary safety endpoint, with vaccine-related adverse events restricted to grade 1. Vaccine-induced immune responses were observed in 93.3% of patients across multiple MHC alleles. Three-year progression-free and death-free rates were 0.63 and 0.84, respectively. Patients with immune responses showed a two-year progression-free rate of 0.82. Two patients without an immune response showed tumour progression within two years of first diagnosis. A mutation-specificity score that incorporates the duration and level of vaccine-induced IDH1(R132H)-specific T cell responses was associated with intratumoral presentation of the IDH1(R132H) neoantigen in pre-treatment tumour tissue. There was a high frequency of pseudoprogression, which indicates intratumoral inflammatory reactions. Pseudoprogression was associated with increased vaccine-induced peripheral T cell responses. Combined single-cell RNA and T cell receptor sequencing showed that tumour-infiltrating CD40LG⁺ and CXCL13⁺ T helper cell clusters in a patient with pseudoprogression were dominated by a single IDH1(R132H)-reactive T cell receptor.

We screened 44 patients in 7 out of 8 centres that are part of the German Cancer Consortium (DKTK) and/or the Neurooncology Working Group of the German Cancer Society (NOA; Supplementary Table 1). Of these, 33 patients were included in the trial (Extended Data Fig. 1). The reasons for exclusion are listed in Supplementary Table 2. One patient (ID16) was not vaccinated because of an adverse event (fever of unknown origin) before vaccination. Hence, 32 patients were treated and therefore included in the safety dataset (SDS; Fig. 1). Twenty patients in the SDS (62.5%) were male and 12 (37.5%) female, and the mean age was 40.4 ± 8.95 years (mean ± s.d.). The trial population was divided into three treatment groups (TG1–TG3) on the basis of standard of care (SOC) treatment that patients had received before enrollment: radiotherapy alone (RT, TG1), three cycles of chemotherapy with TMZ alone (mono-TMZ, TG2) or combined radiochemotherapy with TMZ (RT + cTMZ, TG3) (Extended Data Fig. 1). Most patients had both radiochemotherapy and TMZ before IDH1 vaccination (*n* = 23, 71.9%); three (9.4%) were treated with TMZ alone and six (18.8%) underwent radiotherapy alone. The average total dose of radiotherapy (*n* = 29) was 59.4 Gy. Out of the 32 patients, 21 (65.6%) had World Health Organization (WHO) grade 3 astrocytoma and 11 (34.4%) had grade 4. The predominant location of the astrocytomas was within the frontal lobes (23/32, 71.9%). In terms of surgery, 17 of the 32 patients (53.1%) had undergone complete resection of the tumour, 12 (37.5%) had undergone subtotal resection, and 3 (9.4%) had undergone a biopsy only. For all astrocytoma tissues with sufficient material (24 of 32; 75.0%), the methylation subclass was defined retrospectively. Low-grade methylation accounted for 14 of these 24 astrocytomas (58.3%), and the remaining 10 (41.7%) were methylation class high grade (Fig. 1, Supplementary Table 3).

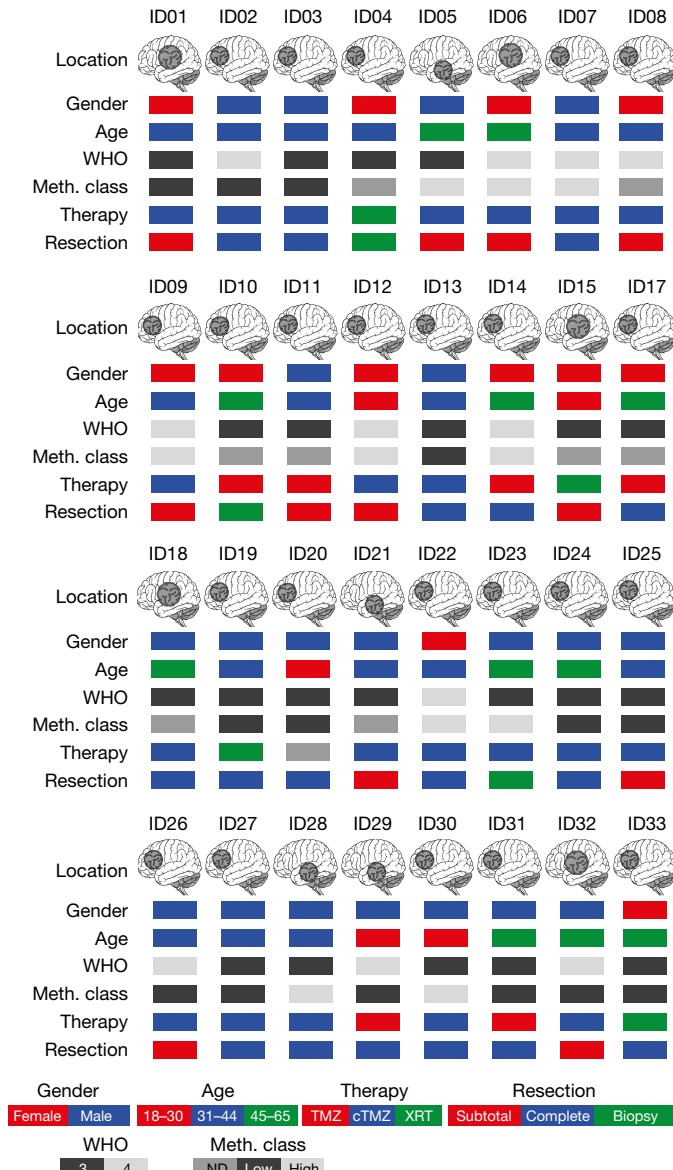

**Fig. 1 | Patient characteristics at baseline and SOC treatment.** cTMZ, concomitant TMZ (75 mg m$^{-2}$ body surface area (BSA)) daily during radiotherapy; TMZ, monotherapy with TMZ (three cycles); XRT, radiotherapy (30 × 2 Gy, if not specified otherwise in Supplementary Table 3); Low, low grade methylation (Meth.) class; High, high grade methylation class; ND, not determined; WHO, WHO grade of tumour. $n$ = 32 patients. Brain illustration taken from Adobe Stock Standard under License ID 222738500.

Two patients (ID19, ID21) were enrolled but could not be evaluated for immunogenicity testing and were therefore excluded from the immunogenicity analysis. Thirty out of the 32 patients in the SDS (93.8%) and 28 of the 30 patients in the immunogenicity dataset (IDS; 93.3%) reached the end of treatment (EOT). The maximum treatment duration was 23 weeks. The median follow-up time (as of June 2020) was 46.9 months (95% confidence interval (CI): 45.2–49.2 months) for the SDS and 47.1 (45.2–49.2) months for the IDS.

## IDH1-vac is safe and immunogenic

The SDS comprised 249 vaccines administered to 32 patients. Twenty-nine out of 32 patients in the SDS (90.6%) and 27 out of 30 patients in the IDS (90.0%) received all 8 vaccinations; one patient received 7, one received 6, and one received 4 vaccinations. The duration of treatment in the SDS ranged from 44 to 162 days (median, 155 days) and the duration of observation ranged from 153 to 484 days (median 376 days). Quality control demonstrated that all vaccines contained 300 ± 30 µg peptide, and were sterile and free from endotoxins. No regime-limiting toxicity (RLT) was observed. The overall serum cytokine profile was indicative of an adverse cytokine release in response to IDH1-vac (Extended Data Fig. 2). Twenty-nine of the 32 patients (90.6%) had treatment-related adverse events, none of which was severe. One patient (3.1%) had treatment-related serious adverse events, and one patient (3.1%) temporarily discontinued the study drug due to treatment-related adverse events (Supplementary Tables 4, 5). Twenty-one (65.6%; 95% CI 46.81–81.43%) and 15 (46.9%; 95% CI 29.09–65.26%) of the adverse events classified as possibly related to IDH1-vac were local administration site conditions (injection site induration or erythema, respectively). Of the 30 patients in the IDS, 28 (93.3%; 95% CI 77.93–99.18%) displayed IDH1-vac-induced immune responses (Fig. 2a, b). IDH1-vac-induced T cell immune responses were observed in 26 of 30 patients and B cell immune responses in 28 of 30 patients across multiple human leukocyte antigen (HLA) alleles; these responses did not correlate with in vitro HLA affinities of the IDH1(R132H) peptide (Extended Data Fig. 3, Supplementary Table 6). Two patients (6.7%) developed neither T cell nor B cell immune responses (Extended Data Fig. 3). To incorporate the duration and level of IDH1-vac-induced T cell immune responses specifically to IDH1(R132H), we established an explorative mutation-specificity score (MSS; Supplementary Table 7, Extended Data Figs. 4, 5). Flow cytometric effector sub-phenotyping of peripheral IDH1-vac-induced T cells from available patient samples with high MSSs showed predominant tumour necrosis factor (TNF), interferon-γ (IFNγ), and interleukin-17 (IL-17) cytokine production by T helper (T$_H$) cells upon in vitro re-stimulation with IDH1(R132H), which indicates the involvement of T$_H$1 and T$_H$17 subtypes of T$_H$ cells (Fig. 2c, Extended Data Fig. 4). Neither IL-10 production by regulatory T cells nor TNF or IFNγ production by cytotoxic T cells was observed (Fig. 2c). Moreover, the MSS was associated with intratumoral IDH1(R132H) antigen presentation in pre-treatment tumour tissue, as assessed by an in situ MHCII–IDH1(R132H) proximity ligation assay (PLA)[5] (Fig. 2d, Extended Data Fig. 5 and Supplementary Table 8).

## Efficacy of the IDH1-vac

The overall response rate was 84.4% (95% CI 67.21–94.72%, 27 of 32 patients) of the SDS, corresponding to 86.7% (95% CI 69.28–96.24%, 26 of 30 patients) of the IDS at the end of study (EOS; Fig. 3a, Extended Data Fig. 6). In followup analyses of the SDS, three-year progression-free and death-free rates were 0.63 (95% CI 0.44–0.77) and 0.84 (95% CI 0.67–0.93), respectively. Two patients of the IDS (ID05 and ID30) who did not mount an IDH1-vac induced immune response (Extended Data Fig. 3) showed progression within two years, compared to patients with immune responses (two-year progression-free rate of 0.82 (95% CI 0.623–0.921) in patients with immune responses; Fig. 3b).

## IDH1-specific T cell responses and pseudoprogression

In the SDS, pseudoprogression (PsPD) occurred in 12 of 32 patients (37.5%) compared to 10 of 60 (16.7%) in a molecularly matched control cohort (Supplementary Table 3). Contrast-enhancing PsPD diagnosed by brain imaging is indicative of intratumoral inflammatory reactions (Fig. 3c). There was no apparent association with age, extent of resection, SOC treatment, or WHO grade. The longer median observation period (7.3 years) in the matched cohort caused a bias towards the detection of more PsPD. In NOA16, PsPD was associated with the onset of peripheral IDH1-vac-induced immune responses (Extended Data Fig. 7) and was restricted to patients with transient or sustained T cell immune responses; we did not detect PsPD in non-responder patients (Fig. 3d).

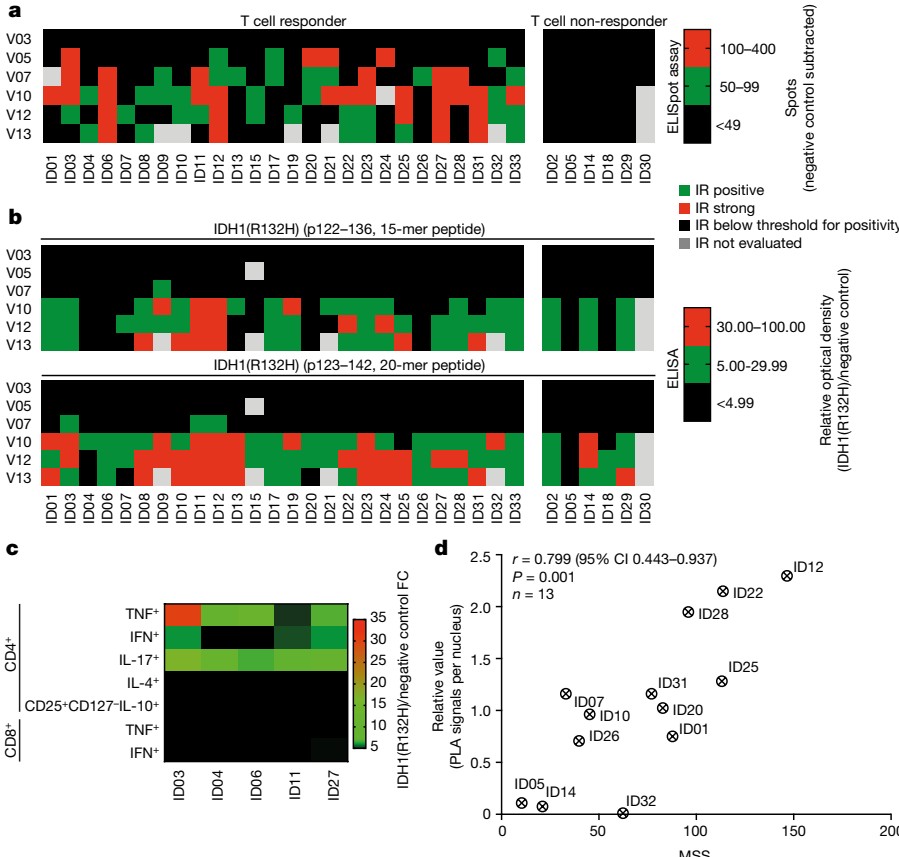

**Fig. 2 | Cellular and humoral immunogenicity of IDH1-vac.**
**a**, **b**, Semi-quantitative analysis of T cell (**a**) and B cell (**b**) immune responses (IR) in all patients in the IDS measured by IFNγ enzyme-linked immunosorbent spot (ELISpot) assay (**a**) or IDH1 peptide enzyme-linked immunosorbent assay (ELISA) (**b**) ($n = 30$ patients). Patients are classified as T cell responders ($n = 24$ patients) and non-responders ($n = 6$ patients) on the basis of specific spot count cut-off of 50 as defined in the study protocol. Response for each visit (V) is shown. **c**, Flow cytometric effector phenotyping of peripheral IDH1-vac-induced T cells (single live $CD3^+$ cells) from available patient samples with high

MSS ($n = 5$ patient samples). Relative values after re-stimulation with IDH1(R132H) peptide compared to negative control peptide (myelin oligodendrocyte glycoprotein; MOG) are shown. FC, fold-change. Gating strategy is shown in Extended Data Fig. 4c. **d**, Correlation of intratumoral IDH1(R132H) peptide presentation at baseline (quantified by PLA signal) with the magnitude and sustainability of specific peripheral T cell responses (quantified by the MSS; see Extended Data Fig. 5). *r*, Pearson correlation coefficient. Patient ID numbers are shown in **a**–**d**.

Patients with PsPD had higher maximal levels of peripheral IDH1-vac induced T cell immune responses than patients who had progressive disease (Fig. 3e). Retrospective assessment of prognostic molecular markers in pre-treatment astrocytoma tissues (Supplementary Table 9, Extended Data Fig. 8) enabled further subgrouping of 24 out of 32 (75.0%) patients in the SDS. PsPD was not associated with any of the assessed tumour-intrinsic molecular markers, such as copy number variation load (CNV-L), methylation class, *CDKN2A* or *CDKN2B* deletion status, frequencies of peripheral immune cell subsets, or alterations in top peripheral T cell clonotypes (Supplementary Table 9, Extended Data Figs. 7, 9, 10). During followup, four out of ten patients (40%) with methylation class high grade glioma experienced progressive disease. Of these, patients with an MSS that stayed below median had a 2-year progression-free rate of 0.4 (95% CI 0.052–0.753) compared to 0.8 (95% CI 0.204–0.969) for patients with an MSS that reached above median, despite an equal distribution of unfavourable molecular markers (Fig. 3f, g). Seven out of 12 (58.3%) of the patients with PsPD, including patient ID08, still have stable disease with a median followup time of 53.1 months (95% CI 45.8–58.2 months).

## Specific T cell receptor in PsPD

Among patients with PsPD, only patient ID08 underwent resection of the lesion (Supplementary Table 8). An ex vivo IFNγ ELISpot assay

with lesion-infiltrating leukocytes (LILs) showed IDH1(R132H)-reactive T cells (Fig. 4a). On the basis of preclinical data[4,5,8] and the observations that neither actively cytotoxic cytokine-producing ex vivo $CD8^+$ LILs nor selected $CD8^+$ T cell clonotype-retrieved T cell receptor (TCR)-transgenic cells reacted to IDH1(R132H) (Extended Data Fig. 11), we focused on $CD4^+$ T cells. Single-cell RNA sequencing (scRNA-seq) identified three clusters of $CD4^+$ T cells within the PsPD lesion of patient ID08: regulatory T cells, activated $CD40LG^+CD4^+$ T cells, and $CXCL13^+CD4^+$ T cells (Fig. 4b, c, Extended Data Fig. 12). $CXCL13^+CD4^+$ T cells have been reported to be important for antitumour immunity[9]. By combining scRNA-seq and TCR sequencing, we found that both the $CD40LG^+CD4^+$ and $CXCL13^+CD4^+$ T cell clusters were dominated by one TCR (TCR14; Fig. 4d). In total, TCR14 was the fourth most abundant TCR within the $CD4^+$ single T cell repertoire, whereas the top three abundant $CD4^+$ TCRs (TCR11–13) were expressed on regulatory T cells that largely lacked TCR14. TCR14 was enriched 50.6-fold in the PsPD lesion of patient ID08 compared to peripheral blood of this patient after administration of IDH1-vac (Extended Data Figs. 10, 12). Transgenic TCR expression in a TCR-deficient human T cell line co-cultured with autologous antigen-presenting cells from patient ID08 showed that TCR14 reacted to IDH1(R132H) (Fig. 4e, Extended Data Fig. 12). These results indicate that IDH1-vac induced clonal expansion of IDH1(R132H)-specific $T_H$ cells that infiltrated into the resected lesion.

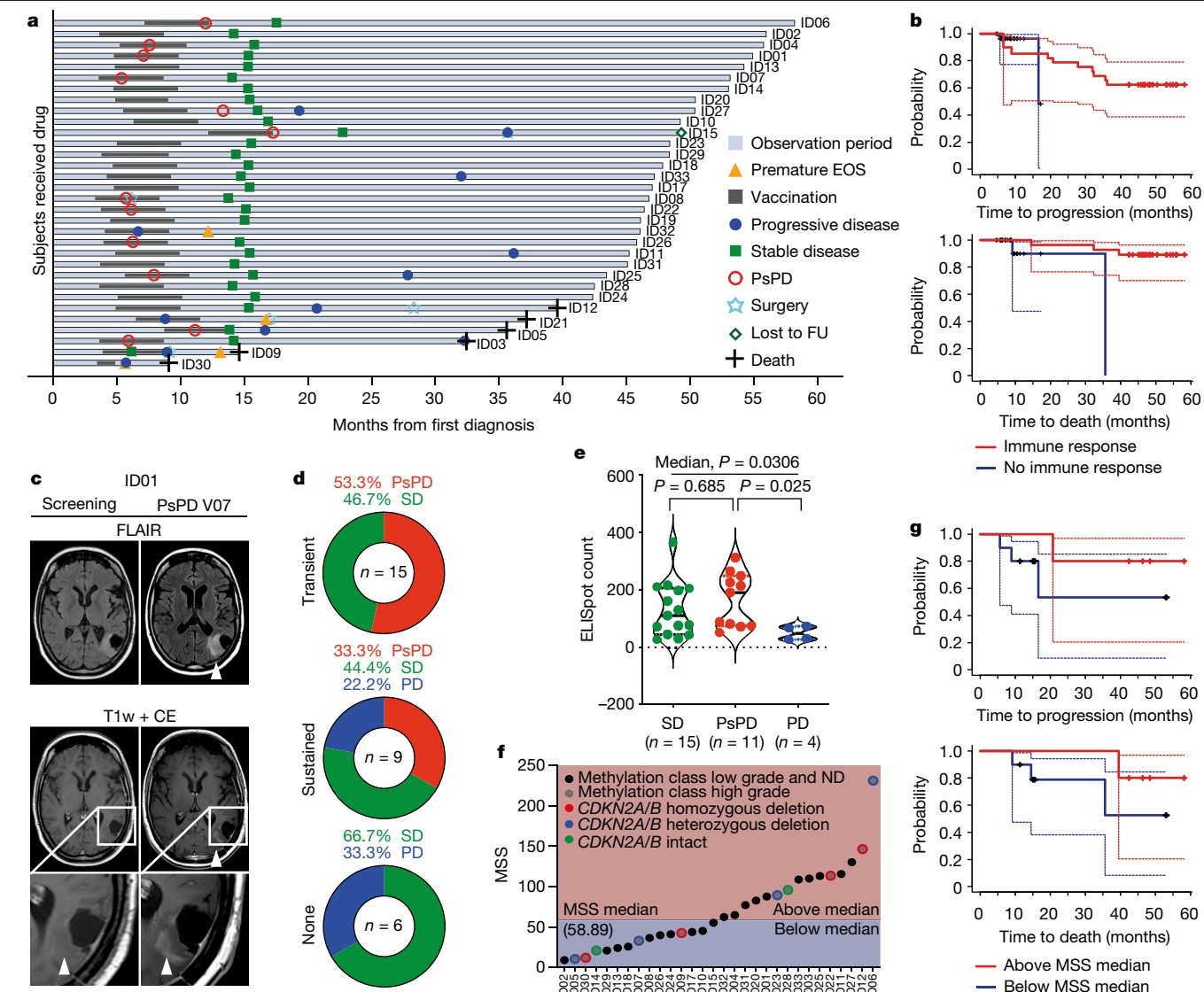

**Fig. 3 | Efficacy of IDH1-vac, pseudoprogression and T cell response.**
**a**, Swimmer plot depicting disease progression and interventions for each patient in the SDS (*n* = 32 patients). **b**, Simon and Makuch plot of overall and progression-free survival probabilities according to the time-dependent covariate IDH1-vac-induced immune response in the IDS (*n* = 30 patients). *x*-axes show time since first diagnosis. **c**, Exemplary MRI fluid-attenuated inversion recovery (FLAIR) and T1-weighted with contrast enhancement (CE) sequences of PsPD of patient ID01 at visit 12 compared to clinical screening MRI. **d**, Frequencies of PsPD, stable disease (SD), and progressive disease (PD) according to T cell response types for patients in the IDS. For definition of transient and sustained responses, see Methods. **e**, Magnitude of best T cell

response defined by maximum specific ELISpot count with negative control subtracted according to disease progression. Individual values, median (solid lines), and quartiles (dotted lines) are shown in violin plots. SD, 95% CI 80–183; PsPD, 95% CI 103–228; PD, 95% CI 11–88. Two-sided Kruskal–Wallis test, Dunn's multiple comparison. **f**, Mutation-specificity scores and molecular profile of each patient in the IDS (*n* = 30 patients). Methylation class low grade or ND, *n* = 20; methylation class high grade, *n* = 10, of which *CDKN2A/B*−/− *n* = 4, *CDKN2A/B*+/− *n* = 4, and *CDKN2A/B*+/+ *n* = 2 patients. **g**, Simon and Makuch plot of overall and progression-free survival probabilities according to the time-dependent covariate MSS in molecularly defined methylation class high grade gliomas. *n* = 10 patients. *x*-axes show time since first diagnosis.

## Conclusions

NOA16 met its primary endpoints by demonstrating the safety and immunogenicity of IDH1-vac in patients with newly diagnosed WHO grade 3 and 4 IDH1(R132H)+ astrocytomas without further positive prognostic factors. Immunogenicity, irrespective of HLA type, and the high rate of PsPD warrant further clinical investigation of IDH1-vac. Patients who did not mount an IDH1-vac induced immune response showed reduced efficacy of the vaccine and disease progression within two years (Extended Data Fig. 3) compared to patients who did mount an immune response (Fig. 3b). IDH1-vac was immunogenic across multiple HLA alleles, supporting the concept of promiscuity of presentation on MHCII[4] and justifying patient inclusion independent of HLA

alleles. To characterize the specificity and dynamics of vaccine-induced peripheral immune responses, we used deep TCR sequencing from most of the patient samples, in addition to central imaging review, molecular pathology and immune monitoring. Single-cell sequencing of T cells from post-vaccine peripheral blood and a tissue sample provided important insights into vaccine-induced systemic and local immune responses and the underlying biological mechanisms of vaccine-induced PsPD, which was associated with a favourable clinical course in some patients. Although this study provides strong circumstantial evidence of de novo induction of cytotoxic T cell responses by IDH1(R132H)-reactive T$_H$ cells within the CNS[10] (Fig. 2c, Extended Data Figs. 11, 12), further functional investigations using trial tissues are required. The high frequency of PsPD in NOA16 participants compared

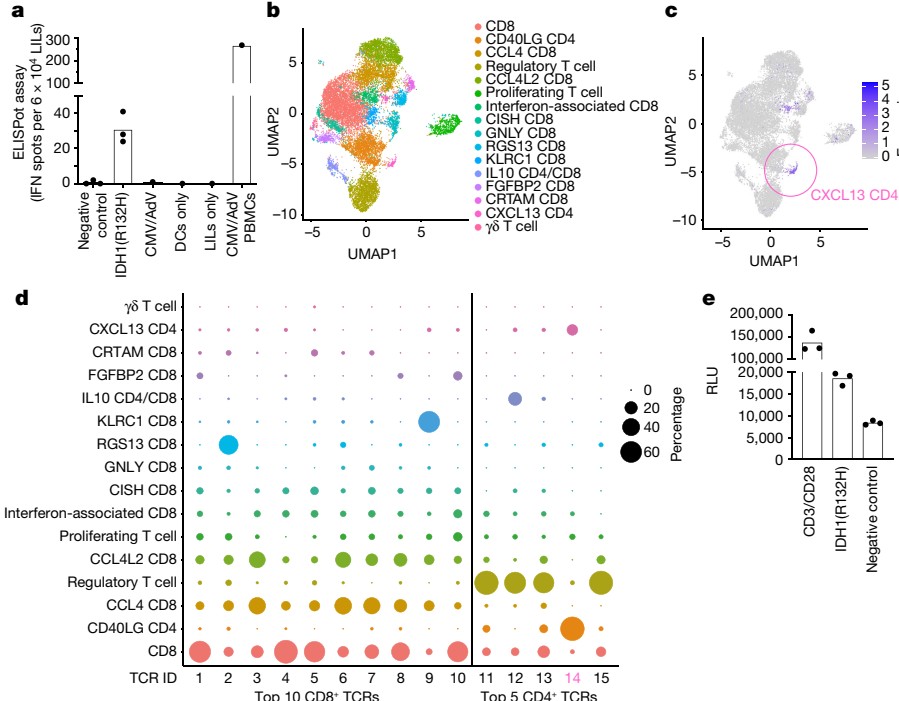

**Fig. 4 | Molecular T cell phenotype of IDH1-vac-associated PsPD. a**, IFNγ ELISpot counts of LILs from PsPD of patient ID08 at visit 07 after ex vivo stimulation with indicated reagents (see Methods). Peripheral blood mononuclear cells (PBMCs) stimulated with cytomegalovirus and adenovirus (CMV/AdV) peptides were used as positive assay control. Data shown as individual values and the mean of three technical replicates (for negative control, IDH1(R132H)). Technical unicates for CMV/AdV, dendritic cells (DCs) only, LILs only, CMV/AdV PBMCs. **b**, UMAP plot depicting molecular clusters defined by single-cell transcriptome of LILs (*n* = 16,720 cells) from PsPD of patient ID08. **c**, CXCL13 expression in LILs from PsPD of patient ID08 within clusters as in **b**. **d**, Bubble plot mapping top TCR clones in CD4⁺ and CD8⁺ T cells defined by single-cell TCR sequencing onto transcriptomic clusters defined in **b**. **e**, T cell activation measured by luciferase NFAT reporter assay after overexpression of a top-five CD4⁺ TCR (TCR14 in **d**) in human Jurkat T cells and co-culture with peptide-loaded autologous PBMCs. Data depicted as individual values and the mean of three technical replicates. Representative of three independent experiments.

to a molecularly matched cohort and previous reports (3 out of 60 patients, 5.0%)[11] may indicate an intratumoral immune reaction that results in disruption of the blood–brain barrier and contrast enhancement. The Response Assessment in Neuro-Oncology (RANO) criteria applied in this trial consider PsPD to be secondary to radiotherapy or combined radiochemotherapy with TMZ, particularly four weeks after completion of radiotherapy[12], and most trials mandate follow-up imaging to demonstrate true progression in cases of suspected PsPD, according to RANO[13,14]. We excluded patients with suspected PsPD from the NOA16 trial, thereby enriching for patients with IDH1-vac-induced PsPD, but we acknowledge that late PsPD may occur as a result of radiotherapy[15]. Also, late PsPD six months after initiation of immunotherapy may occur, as acknowledged in the immunotherapy RANO (iRANO) criteria[13], which were not defined at the time of initiation of this trial. Notably, the rate of PsPD in NOA16 did not differ when analysed according to iRANO criteria. However, there are limitations to definitive proof of PsPD, even with positron emission tomography (PET) imaging or histologic analysis of re-resection, as no firm criteria exist[16].

NOA16 is based on strong preclinical data[4,6] and the decision to integrate IDH1-vac into the primary treatment of newly diagnosed patients provided a sufficient therapeutic window and allowed us to exploit potential positive immune interactions between SOC and vaccination. While this strategy has been chosen in other trials that have targeted shared[17] or personalized neoepitopes[18,19], NOA16 targeted a shared clonal neoepitope to minimize the risk of immune evasion by clonal selection or spontaneous neoantigen loss[17]. Clonality of neoepitopes is a key determinant of efficacy for immune checkpoint inhibitors across many cancer entities[20]. Gliomas are particularly prone to the development of subclonal mutational events that

contribute to resistance to immune checkpoint inhibitors[21]. Targeting a shared clonal driver mutation in newly diagnosed patients overcomes these limitations[6] and may provide a basis for future trials that target MHCII-restricted clonal shared and personalized neoepitopes in cancer immunotherapy.

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

[1]DKTK (German Cancer Consortium) Clinical Cooperation Unit (CCU) Neuroimmunology and Brain Tumor Immunology, German Cancer Research Center (DKFZ), Heidelberg, Germany. [2]Department of Neurology, Medical Faculty Mannheim, MCTN, University of Heidelberg, Mannheim, Germany. [3]Immune Monitoring Unit, National Center for Tumor Diseases (NCT), Heidelberg, Germany. [4]Neurology Clinic, Heidelberg University Hospital, University of Heidelberg, Heidelberg, Germany. [5]NCT, Heidelberg, Germany. [6]NCT Trial Center, NCT, Heidelberg, Germany. [7]Department of Neuroradiology, Heidelberg University Hospital, University of Heidelberg, Heidelberg, Germany. [8]DKTK CCU Neuropathology, DKFZ, Heidelberg, Germany. [9]Department of Neuropathology, Heidelberg University Hospital, University of Heidelberg, Heidelberg, Germany. [10]Immunitrack, Copenhagen, Denmark. [11]DKMS Life Science Lab GmbH, Dresden, Germany. [12]Department of Internal Medicine V, Heidelberg University Hospital, University of Heidelberg, Heidelberg, Germany. [13]Department of Neurosurgery, University of Freiburg, Freiburg, Germany. [14]Department of Medical Oncology, West German Cancer Center, University Hospital Essen, University of Duisburg-Essen, Essen, Germany. [15]Department of Neurosurgery, Charité Medical Center, University of Berlin, Berlin, Germany. [16]Department of Neurosurgery, Carl Gustav Carus University Hospital, University of Dresden, Dresden, Germany. [17]Institute of Cell Biology, Department of Immunology, University of Tübingen, Tübingen, Germany. [18]Department of Neurology, University of Tübingen, Tübingen, Germany. [19]Dr. Senckenberg Institute of Neurooncology, Frankfurt, Germany. [20]DKTK CCU Neurooncology, DKFZ, Heidelberg, Germany. [✉]e-mail: m.platten@dkfz.de

## Methods

### Patients and trial design

NOA16 was a non-controlled, open-label, single-arm, multicentre, first-in-humans phase I trial to assess the safety, tolerability and immunogenicity of eight repeated doses of IDH1-vac in patients with IDH1(R132H)[+], non-1p/19q co-deleted, ATRX[−] WHO grade 3 and 4 gliomas. The study ran from May 2015 to November 2018 at seven trial centres in Germany (Supplementary Table 1). Follow-up to evaluate the duration of response, survival, and late adverse events is ongoing. The study was approved by the national regulatory authority (Paul-Ehrlich Institut) and the institutional review board (Ethikkommission) at each study site, namely: Ethikkommission der Medizinischen Fakultät Heidelberg (Heidelberg), Ethik-Kommission Albert-Ludwigs-Universität Freiburg (Freiburg), Ethik-Kommission des Landes Berlin (Berlin), Ethik-Kommission der Medizinischen Fakultät der Universität Duisburg-Essen (Essen), Ethik-Kommission der Medizinischen Fakultät "Carl Gustav Carus" (Dresden), Ethik-kommission des Fachbereichs Medizin der Goethe-Universität Frankfurt am Main (Frankfurt), Ethikkommission der Medizinischen Fakultät der Ludwig-Maximilians-Universität München (Munich), Ethik-Kommission an der Medizinischen Fakultät der Eberhard-Karls-Universität und am Universitätsklinikum Tübingen (Tübingen). The study was conducted in accordance with the Good Clinical Practice guidelines of the International Conference on Harmonisation. All participants provided written signed informed consent. We complied with all relevant ethical regulations. The trial population comprised three treatment groups (TGs) based on the SOC treatment that patients had received before enrollment: radiotherapy alone (RT, TG1), three cycles of chemotherapy with TMZ alone (mono-TMZ, TG2) or combined radiochemotherapy with TMZ (RT + cTMZ, TG3). In TG1, vaccination was done alone starting 4–6 weeks after radiotherapy. In TG2 and TG3, vaccination was done in parallel with TMZ starting on day 10 of the fourth cycle of the TMZ monotherapy (TG2) or on day 10 of the first adjuvant (a)TMZ cycle after concomitant radiotherapy (TG3). Treatment consisted of eight vaccinations with IDH1-vac in weeks 1, 3, 5, 7, 11, 15, 19 and 23 (visits (V) 03–10; Extended Data Fig. 1b). For immunogenicity assessment, peripheral T cell and B cell immune responses were assessed at six time points: V03 (baseline), V05, V07, V10, V12, and V13 (Extended Data Fig. 1b). Eligibility criteria included the presence of a histologically confirmed IDH1(R132H)[+] glioma (with or without measurable residual tumour after resection or biopsy) with absence of chromosomal 1p/19q co-deletion and loss of nuclear ATRX expression in the tumour tissue, thus limiting inclusion in this first-in-humans trial to the subgroup of molecular astrocytoma without positive prognostic factors[22]. Exclusion criteria included concomitant treatment with dexamethasone (or equivalent) >2 mg/day, Karnofsky performance status (KPS) < 70, and progressive (including PsPD[14]) or recurrent disease after SOC. The matched control cohort was built from patients treated at the centre in Heidelberg outside the trial between 2007 and 2018 with sufficient clinical and MRI information available to assess PsPD. Matching was done according to the first treatment phase of a histologically confirmed IDH1(R132H)[+] glioma (with or without measurable residual tumour after complete or partial resection or biopsy) without 1p/19q co-deletion or loss of nuclear ATRX expression in the tumour tissue, and according to WHO grade 3 or 4 as well as frequency of treatment adaptions (RT + cTMZ versus mono-TMZ or RT; Supplementary Table 3). No statistical methods were used to predetermine sample size. Sample size estimation was primarily based on the accuracy requirements for the primary endpoint immune response (responder rate) to the IDH1 peptide vaccine. Sample size was adjusted for non-evaluable patients. It was estimated that 70% of patients who would be evaluable for immunogenicity testing would be evaluable for all time points. Because 21 patients were sufficient for immunogenicity testing with all time points, 30 evaluable patients had to be enrolled. Owing to an expected dropout rate of 20% (due to progression or other reasons), 39 patients had to be recruited. All patients received the trial-related intervention; the trial was not randomized and investigators were not blinded concerning trial related intervention during experiments and outcome assessment.

### IDH1 vaccination

IDH1-vac consisted of 300 μg of an IDH1(R132H) 20-mer peptide (p123–142) manufactured by the GMP facility of the University of Tübingen, Germany and emulsified in Montanide (ISA50) as described earlier[23] by the GMP core facility at the University Hospital Heidelberg, Germany, a maximum of one day in advance. It was administered subcutaneously in combination with topical imiquimod (5%, Aldara). Quality controls for content, sterility and absence of endotoxin were performed for each emulsion at Labour LS s.e. & Co. KG, Germany.

### Endpoints

The primary endpoints were safety and immunogenicity. The safety endpoint was the RLT, which was defined as one of the following that was related to IDH1-vac administration: any injection site reaction of National Cancer Institute Common Terminology Criteria for Adverse Events (CTCAE) version 4.0 grade 4; any injection site reaction of CTCAE grade 3 that persisted after two weeks; any other hypersensitivity, anaphylaxis or local allergic reaction of at least CTCAE grade 3; brain oedema (CTCAE grade 4); autoimmunity of CTCAE grade 3 or more; CTCAE grade 3 or more toxicity to organs other than the bone marrow, but excluding grade 3 nausea, grade 3 or 4 vomiting in patients who had not received optimal treatment with anti-emetics, grade 3 or 4 diarrhoea in patients who had not received optimal treatment with anti-diarrheals, and grade 3 fatigue; and death. Adverse events were counted as treatment-related if the relationship to treatment was 'certain', 'related', 'probable', 'possible', or not reported. For safety assessment, patients were medically reviewed at each visit. To exclude unexpected IDH1-vac-induced immunological tolerance against IDH1(R132H), shortening of progression-free survival (PFS), defined as an observed decrease in the estimated 12-months PFS rate of at least 10% compared to the anticipated value of 70.7% derived from previous studies, was defined as a safety criterion for early trial termination. The safety analysis was based on all enrolled patients who received one or more administration(s) of IDH1-vac. The immunogenicity endpoint was defined as the presence of an IDH1(R132H)-specific T cell and/or antibody response at any time point during the trial. IDH1(R132H)-specific T cell and antibody responses were measured on PBMCs using IFNγ ELISpot and on serum using peptide-coated ELISA, respectively. For IFNγ ELISpot, a cut-off of 50 IFNγ spots after subtraction of negative control was defined as positive. For ELISA, the cut-off for positivity was defined as optical density related to negative control ≥5.

### Disease assessment

Disease assessment, including overall response rate and diagnosis of PsPD, was performed using standardized three-monthly MRI according to the RANO criteria by central neuroradiology review[14]. In NOA16 and the molecularly matched control cohort, PsPD, which may mainly indicate an intratumoral inflammatory reaction[24], was defined as an increase in the size of the tumour on T2-FLAIR MRI sequences and/or the novel appearance or enlargement of contrast-enhancing lesions followed by stabilization or regression on follow-up MRI up to three months after initiation of SOC and/or immunotherapy[14].

### Preparation of peptides for analyses

Lyophilized peptides were reconstituted in 100% DMSO and diluted to a final concentration of 10 mg ml[−1] with aqua ad iniectabilia (Braun). The final DMSO concentration was 10%.

## Isolation of serum

Serum tubes were kept standing upright at room temperature for 15 min before isolation. Serum tubes were centrifuged at 1,000$g$ for 10 min at room temperature. Supernatant was aliquoted on ice and frozen at −80 °C.

## Isolation of PBMCs

PBMCs were isolated from heparinized blood from patients with glioma by density-gradient centrifugation (800$g$ without brake at room temperature) by loading onto Biocoll Separation Solution (Biochrom) after dilution with phosphate-buffered saline (PBS) and using Leucosep tubes (Greiner Bio-One). PBMCs were frozen in 50% freezing medium A (60% X-Vivo 20, 40% fetal calf serum (FCS)) and 50% medium B (80% FCS, 20% DMSO) and stored in liquid nitrogen at −140 °C until analysis.

## Isolation of LILs

Lesion tissue was dissected into small pieces (2 × 2 mm) and transferred into 24-well tissue culture-treated plates at three pieces per well in 2 ml human tumour-invading lymphocyte (TIL) medium (RPMI1640 (Pan Biotec) with 10% human serum (Sigma Aldrich), 2 mM L-glutamine, 1.25 μg/ml amphotericin B (both Gibco), 1,000 U/ml IL-2 (Proleukin)) containing 30 ng/ml anti-human CD3 (clone OKT-3, eBioscience). Medium was exchanged every 2–3 days and tissue pieces removed on day 7. LILs that migrated out of the tumour into the medium were further expanded until day 14 and cryopreserved as above.

## Generation of patient REP cells

To enable HLA-autologous testing for antigen-specific reactivity of LILs and TCR-transgenic cells, patient-autologous rapidly expanded PBMCs (REP cells), which express high levels of MHC molecules and can serve as antigen-presenting cells (APCs), were generated. PBMCs ($1 \times 10^5$) were co-cultured in a high-density culture with $3 \times 10^7$ irradiated (40 Gy) feeder cells (PBMCs from non-autologous donors) in X-vivo15 medium supplemented with 2% human AB serum (Sigma-Aldrich) and 30 ng/ml OKT-3 antibody (Invitrogen) in T-25 flasks in 25 ml total volume. After 24 h, cells were supplemented with 300 IU/ml hIL-2. Medium was replaced every 5 days with hIL-2 supplementation and cells were split as needed. Cells were collected after 14 days of co-culture and cryopreserved.

## IFNγ ELISpot of PBMCs

ELISpot white-bottom multiscreen HTS plates (MSIPS4W10, Millipore) were coated with anti-human IFNγ (1-D1K, Mabtech) and blocked with X-Vivo-20 (Lonza) containing 2% human albumin (HA). PBMCs were thawed, rested overnight in X-Vivo medium and seeded at $4 \times 10^5$ cells per well and stimulated with 2 μg peptides per well in 100 μl volume. PBMCs were stimulated with IDH1(R132H) (p123–142), wild-type IDH1 (p123–142), or MOG (p35–55) at equal concentrations or with peptide diluent aqua ad iniectabilia (Braun) with 10% DMSO (vehicle) at equal volume as negative controls, or with 1 μg staphylococcal enterotoxin B (Sigma-Aldrich) per well and 0.05 μg CMV with 0.05 μg AdV per well (both in 100 μl volume) as positive controls. After 40 h, IFNγ-producing cells were detected with biotinylated anti-human IFNγ antibodies (7-B6-1), streptavidin-ALP (both Mabtech) and ALP colour development buffer (Bio-Rad) and quantified using an ImmunoSpot Analyzer (Cellular Technology Ltd). Quality control was performed and reviewed by a second person. For categorization of T cell responses, transient T cell responses were defined as a spot count above 50 followed by a spot count of less than 50 at EOS. Sustained T cell responses were defined as a spot count above 50 followed by a spot count of more than 50 at EOS.

## IFNγ ELISpot of LILs

To generate dendritic cells (DCs) to serve as antigen-presenting cells, autologous patient PBMCs were thawed in X-Vivo-20 medium and plated on tissue-culture-treated plates at a density of $5 \times 10^6$ cells per ml for 1 h. The supernatant was removed and adherent monocytes were differentiated into DCs by culturing in X-Vivo-20 medium containing 500 U/ml hIL-4 (Miltenyi) and 560 U/ml human granulocyte-macrophage colony-stimulating factor (hGM-CSF) (Genzyme) for 7 days. DCs were collected and purified using magnetic-activated cell sorting (MACS). Anti-CD56 antibodies coupled to pan mouse IgG Dynabeads, CD19 pan B Dynabeads and CD3 Dynabeads (all Invitrogen) were used to remove contaminating cell populations according to the manufacturer's protocol. To enrich LILs for antigen-reactive T cells, DCs were seeded at a density of $2 \times 10^5$ cells per ml in RPMI1640 medium containing 10% AB serum, 100 U/ml penicillin, and 100 μg/ml streptomycin, and loaded with 10 μg/ml IDH1(R132H) (p123–142) for 4 h. They were then co-cultured with LILs, which had been thawed and rested overnight in X-VIVO-20 medium, at a ratio of 1:5 (DCs:LILs). For proliferation of T cells, from day 3 onwards, co-culture medium was supplemented with 40 U/ml IL-2 (Proleukin) and 20 ng/ml IL-7 (Peprotech) and refreshed every 2 to 4 days. LILs were collected after 24 days of co-culture, rested overnight in RPMI1640 medium containing 10% AB serum, 100 U/ml penicillin, and 100 μg/ml streptomycin, and used for ELISpot in co-culture with freshly isolated autologous DCs as above, which had been loaded with 2 μg/100 μl IDH1(R132H) peptide (p123–142) or MOG peptide (p35–55) as negative control overnight in the same medium, at a ratio of 1:6 ($1 \times 10^4$ DCs:$6 \times 10^4$ LILs) for 40 h. ELISpot was performed as described above.

## Flow cytometry

For peripheral immune monitoring, $3 \times 10^5$ PBMCs were stained with the following antibodies targeting surface proteins: anti-CD3-FITC (clone UCHT1, cat # 300452, 1:100), anti-CD4-Alexa Fluor700 (clone RPA-T4, cat # 300526, 1:100), anti-CD8-PerCP (clone RPA-T8, cat # 301030, 1:100), anti-CD11b-BV510 (clone M1/70, cat # 101263, 1:20), anti-HLA-DR-PE-Cy7 (clone L243, cat # 307616, 1:50), anti-CD14-BV711 (clone M5E2, cat # 301838, 1:100), anti-CD16-PE/Dazzle594 (clone 3G8, cat # 302054, 1:10), anti-CD25-BV605 (clone BC96, cat # 302632, 1:20), anti-CD33-APC (clone P67.6, cat # 366606, 1:50), and anti-CD127-BV421 (clone A019D5, cat # 351310, 1:20) (all BioLegend); and fixable viability dye-eFluor780 (1:1,000, Invitrogen), followed by intracellular staining with anti-FOXP3-PE (clone 206D, cat # 320108, 1:100, BioLegend) using the Fixation and Permeabilization Buffer Set (ebioscience). Antibody amounts were titrated previously. In all experiments, corresponding fluorescence minus one (FMO) controls were used (Extended Data Fig. 9). As many events as possible were measured on an Attune NxT Flow Cytometer using Attune Nxt software version 2.7 (ThermoFisher Scientific).

For analysis of IDH1(R132H)-reactive T cell subsets, we performed an ex vivo peptide recall assay. PBMCs were thawed, rested for 4 h in X-Vivo 20 medium, and seeded into 96-well U-bottom plates. PBMCs ($1.5–2 \times 10^6$) were stimulated with 2 μg peptide per well using IDH1(R132H) (p123–142), MOG (p35–55) as negative control, or CEFT peptide pool (0.05 μg/ml per peptide, jpt) as positive control for 3 h before adding 10 μg/ml brefeldin A (Sigma-Aldrich, order no. B6542) and 1× GolgiStop (BD Bioscience). Cells were incubated for an additional 12 h and subsequently stained with the following surface antibodies: anti-CD3-BV510 (clone HIT3a, cat # 564713, 1:20), anti-CD4-BV605 (clone SK3, cat # 566908, 1:50), anti-CD8-APC-H7 (clone SK1, cat # 560179, 1:10) (panels 1 and 2), anti-CD25-BV711 (clone 2A3, cat # 563159, 1:10), and anti-CD127-FITC (clone HIL-7R-M21, cat # 560549, 1:2.5) (panel 2) (all BD Biosciences); and fixable viability dye-APC-R700 (1:1,000, Invitrogen), followed by intracellular staining with anti-IFNγ-BV421 (clone 4S.B3, cat # 564791, 1: 20, BD Biosciences), anti-TNF-APC (clone MAb11, cat # 502912, 1:20, Biolegend), anti-IL17-PE (clone N49-653, cat # 560486, 1:5), and anti-IL4-PerCP-Cy5.5 (clone 8D4-8, cat # 561234, 1:20) (panel 1), or anti-FOXP3-PE (clone 259D/C7, cat # 560046, 1:5) and anti-IL10-APC (clone JES3-19F1, cat # 554707, 1:50) (all BD Biosciences), using the Foxp3/Transcription Factor Staining Buffer Set (ebioscience).

Antibody amounts were titrated previously or used according to manufacturer's instructions, and scaled up according to cell numbers at time of seeding. In all experiments, corresponding FMO controls were used (Extended Data Fig. 4). As many events as possible were measured on a Lyric Flow Cytometer (BD Bioscience) using BD FACSuite sotware version 1.3.

For fluorescence-activated cell sorting (FACS) of LILs, patient tissue was dissected into small pieces, transferred to HBSS (Sigma Aldrich) and strained successively through 100-µm, 70-µm and 40-µm cell strainers with intermittent washes with HBSS to obtain a single-cell suspension. Cells were stained with the following antibodies targeting surface proteins: anti-CD45-eFluor450 (clone 2D1, cat # 48-9459-42, 1:50, ebioscience) and anti-CD3-PE (clone HIT3a, cat # 300308, 1:50, BioLegend); and fixable viability dye-eFluor780 (1:1,000, Invitrogen). Cells were gated for lymphocytes, single cells and live cells, and sorted into CD45$^+$CD3$^+$ and CD45$^+$CD3$^-$ cell populations (Extended Data Fig. 12) on a FACSAria IIu with FACSDiva software version 8.0 (BD Biosciences).

For ex vivo testing of the reactivity of CD8$^+$ LILs to IDH1(R132H), cryopreserved LILs expanded from tumour pieces and patient-specific REP cells were thawed in X-vivo 15 medium containing 50 U/ml Benzonase (Sigma Aldrich), and rested for 12 h in X-vivo 15 medium with 2% Human AB serum (Sigma-Aldrich) and 20 IU/ml hIL-2 (Proleukin). REP cells were irradiated (30 Gy), seeded in 96-well U-bottom plates at $1 \times 10^5$ cells per well and loaded with 10 µg/ml IDH1(R132H) (p123–142) or MOG (p35–55) peptide for 2 h. In the meantime, LILs were labelled with CFSE (ThermoFisher) according to the manufacturer's protocol to help distinguish them during flow cytometry, and co-cultured with peptide-loaded REP cells at a 1:1 ratio. After 12 h, 10 µg/ml Brefeldin A was added to the co-culture for an additional 5 h. Positive control cells were stimulated with 20 ng/ml phorbol 12-myristate 13-acetate (PMA) and 1 µg/ml ionomycin (Sigma-Aldrich). Cells were subsequently stained with the following surface antibodies: anti-CD3-BV510 (clone HIT3a, cat # 564713, 1:20, BD Biosciences) and anti-CD8-PerCP-Cy5.5 (clone RPA-T8, cat # 45-0088-42, 1:100, ebioscience); and fixable viability dye-eFluor780 (1:1,000, Invitrogen), followed by intracellular staining with anti-TNF-APC (clone MAb11, cat # 17-7349-82, 1:50) and anti-IFNγ-eFluor450 (clone 4S.B3, cat # 48-7319-42, 1:50) (all ebioscience) using the IC Fixation buffer kit (eBioscience). Corresponding FMO controls were used (Extended Data Fig. 11) and events were measured on a FACSCanto II flow cytometer with FACSDiva software version 9.0 (BD Biosciences).

Data analysis for all experiments was done using FlowJo software v.10.5.0.

## IgG ELISA
ELISA polysorp plates (Nunc) were coated with human IDH1(R132H) and human wild-type IDH1 (p122–136 and p123–142) for patient IgG detection, and with negative control MOG (p35–55) (10 µg per well in PBS). Wells were washed with PBS 0.05% Tween 20, and blocked with 3% FBS in PBS 0.05% Tween 20. The positive control for patient serum was tetanus toxoid (Millipore) with EBNA-1 (RayBiotech) (each 0.5 ng per well). Patient and healthy control sera were obtained from serum tubes by centrifugation. Patient serum was used at the following dilutions: 1:10, 1:100, 1:333, 1:1,000 and 1:3,333. Healthy control serum was used undiluted. Mouse anti-IDH1(R132H) (1:1,000, H09, Dianova) was used as peptide coating control. HRP-conjugated secondary antibodies were sheep anti-mouse IgG-HRP (1:5,000, Amersham) and goat anti-human IgG-Fc-HRP (1:10,000, Bethyl Laboratories, Inc.). The substrate was tetramethylbenzidine (ebioscience) and the reaction was stopped with 1 M H$_2$SO$_4$. Optical density was measured at 450 nm.

## Detection of cytokines in serum
Serum was analysed using multiplex bead technology (Bio-Plex Pro Human Cytokine 27-plex panel, order no. M500KCAFOY, Bio-Rad, Hercules, CA) according to the manufacturer´s instructions. Serum was diluted 1:2. Standard curves were generated by using the reference cytokine sample supplied in the kit and were used to calculate the cytokine concentrations in the samples. Acquisition and data analysis were performed by bio-plex Manager.

## Proximity ligation assay
PLA was performed on baseline paraffin-embedded glioma tissues as described previously[5]. For image acquisition, a nonlinear adjustment (gamma changes) was used for visualization purposes.

## TCRB deep sequencing
Genomic DNA was isolated from patient EDTA blood using the DNeasy Blood and Tissue Kit (Qiagen). TCR beta chain (TCRB) deep sequencing was performed to detect rearranged TCRβ gene sequences using hsTCRB Kit (Adaptive Biotechnologies) according to the manufacturer's protocol. The prepared library was sequenced on an Illumina MiSeq by the Genomics & Proteomics Core Facility, German Cancer Research Center (DKFZ). Data processing (demultiplexing, trimming, gene mapping) was done using the Adaptive Biotechnologies proprietary platform. Data were visualized using the Treemap Visualization package version 2.4.2 (https://cran.r-project.org/web/packages/treemap/index.html). TCRB sequencing data are available at https://clients.adaptive-biotech.com/pub/platten-2021-nature.

## Next-generation HLA typing
Genomic DNA was isolated from patient EDTA blood using the QIAamp DNA Blood Mini Kit (Qiagen). Subsequently, peptide-binding domains were sequenced as described previously[25].

## 850k methylation arrays
850k methylation arrays were performed as described previously[26].

## Panel sequencing
DNA from FFPE tissue was extracted on the Promega Maxwell device (Promega) following the manufacturer's instructions. Extracted DNA was then sheared on a Covaris M220 (Covaris). DNA integrity and fragment size were determined on a Bioanalyzer 2100 (Agilent). Sequencing was performed on a NextSeq 500 instrument (Illumina) with an average coverage of 550-fold[27].

## Single-cell RNA and TCR sequencing
Single-cell capturing and downstream library constructions of FACS-sorted cells were performed using Chromium Single Cell V(D)J Reagent kit v1 chemistry (10x Genomics; PN-1000006, PN-1000020, PN-1000005, PN-120262) according to the manufacturer's protocol. The constructed scVDJ library and scGEX libraries were sequenced on HiSeq2500 rapid and HiSeq4000 platforms (Illumina), respectively.

Single-cell RNA data were processed using cellranger pipeline (version 3.1.0) with GRCh38 genome assembly (version 3.0.0, 10x Genomics) with default setting. The filtered matrices were then analysed using Seurat[28]. Cells with fewer than 2,000 unique molecular identifiers, fewer than 900 genes, and/or more than 10% mitochondrial gene expression were excluded from the analysis. Genes detected in fewer than three cells were excluded. Gene expression was transformed and normalized using regularized negative binomial regression as implemented in sctransform[29]. VDJ genes were removed from the variable genes to prevent clustering of cells on the basis of TCR clones. Highly variable genes were selected using principal component analysis, and 40 principal components were selected on the basis of inflection point in the elbow plot. Cells were clustered using graph-based clustering with Louvain modularity of 0.45 and UMAPs were plotted for visualization. Differential gene expression analysis was performed using MAST[30] to determine the identity of each cluster and highly upregulated genes were used to label each cluster. Clusters with upregulated heat shock proteins and CD3$^-$ cells were excluded and cells were re-normalized

and re-clustered as described above. Single-cell VDJ data were processed similarly using cellranger pipeline. Barcodes of individual top TCRs were then mapped onto single-cell RNA data to determine the distribution of TCR clones in the clusters.

Single-cell sequencing data have been deposited in the NCBI Sequence Read Archive with the accession codes SRR12880623 and SRR12880624.

## TCR cloning

Synthetic alpha and beta VDJ fragments of the variable region of the TCR compatible with BsaI-mediated Golden Gate Assembly cloning were obtained from Twist Biosciences. An S/MAR sequence-bearing expression vector (pSMARTer) that allows extrachromosomal replication of the vector in eukaryotic cells was used and designed to harbour mouse alpha and beta constant TCR regions and a p2a self-cleaving peptide linker to facilitate production of separate alpha and beta polypeptide chains of the TCR. The TCR variable fragments were inserted into the expression vector using a single-step Golden Gate reaction and transformed into NEB5-alpha-competent *Escherichia coli* (NEB). Colonies were screened for the transgene by antibiotic resistance, and an endotoxin-free plasmid was prepared using NucleoBond Extra Maxi EF kit (Macherey-Nagel) for transfection.

## TCR-NFAT reporter assay

The cloned TCR expression vector and a nano-luciferase-based NFAT reporter vector (pDONR, with 4× NFAT-response elements) were delivered into Jurkat Δ76 cells (obtained from TRON gGmbH, authenticated using the Multiplexion STR profiling and compared to normal Jurkat cells, regularly tested for mycoplasma contamination and tested negative at all time points) using electroporation (Neon Transfection system, ThermoFisher Scientific). In brief, $2 \times 10^6$ cells were used per electroporation with Neon 100-μl tips (8 μg TCR expression vector with 5 μg NFAT reporter vector). Cells were prepared according to the manufacturer's protocol; electroporated with 1,325 V, 10 ms, 3 pulses; and transferred to antibiotic-free RPM1 1640 medium containing 10% FCS. Patient-autologous PBMCs or REP cells were used as APCs as indicated and thawed 24 h before co-culture in X-VIVO 15 medium (Lonza) containing 50 U/ml benzonase (Sigma-Aldrich), rested for 6–8 h before seeding into 96-well white-opaque tissue culture-treated plates (Falcon) at $1.5 \times 10^5$ cells per well, and loaded with peptides at a final concentration of 10 μg/ml in a total volume of 150 μl for 16 h. A pool of human IDH1(R132H) peptides (p122–136, p124–138, p126–140) was used. MOG (p35–55) at equal concentrations and PBS + 10% DMSO (vehicle) at equal volume were used as negative controls. Forty-eight hours after electroporation, Jurkat Δ76 cells were collected and co-cultured with peptide-loaded PBMCs for 6 h at a 1:1 ratio. Human T cell TransAct beads (Miltenyi) were used as positive control. Nano-luciferase induction, indicating TCR activation, was assayed using the Nano-Glo Luciferase assay system (Promega) according to the manufacturer's protocol and signal was detected on a PHERAstar FS plate reader (BMG Labtech).

## In vitro HLA affinity analyses

Peptides were synthesized by Genscript and dissolved in DMSO followed by dilution in assay buffer. The final DMSO concentration was 10%. Peptides did not contain cysteines so no reducing agent was added. As positive controls, peptides CLIP (PVSKMRMATPLLMQA), KLAT (HA306–318, YKYVKQNTLKLAT) or PADRE (AKFVAAWTLKAAA) were used. Peptides were titrated in assay buffer (10,000, 1,000, 100, 10, 1, 0.1, 0.01, and 0.001 nM) and recombinant MHC II of different alleles and paralogues was added. After at least 24 h of refolding, solutions were transferred to optiplates AlphaScreen acceptors and donor beads were added. Raw data was imported into Microsoft Excel and deconvoluted. For some peptides the highest concentrations led to a reduction in signal (hooking effect). These datapoints were deleted. Data were imported into GraphPad Prism software version 9.0.0 and analysed by sigmoid curve fitting. All experiments were done in duplicate with good correlation.

## Statistics

For statistical analyses of primary endpoints, two patient analysis populations were defined. The safety population included all enrolled patients who had at least received one dose of IDH1-vac. This was the analysis dataset for evaluating patient characteristics, study administration, efficacy (overall response rate, i.e. stable disease), and safety endpoints (safety dataset, SDS). The immunogenicity population (immunogenicity dataset, IDS) included all patients who could be evaluated for immunogenicity assessment. A patient was defined as evaluable if they had completed the study up to and including V07, had received at least four vaccinations through V07 and had all intended blood samples collected for immune monitoring through V07; or had received at least 6 of 8 vaccinations, and baseline plus at least two further blood samples had been collected for immune monitoring through V12. Non-evaluable patients were replaced for assessment of immunogenicity, except for patients who left the study early owing to RLT. For the primary endpoints (RLT and immune response), summary tables, percentages and exact 95% CIs according to Clopper–Pearson were generated.

All secondary variables were analysed using explorative and mainly descriptive methods using GraphPad Prism software version 9.0.0. For PLA, Pearson correlation coefficient was calculated. For contingency analyses, Fisher's exact test was performed. For multiple comparisons, a Kruskal–Wallis test (KWT) by ranks was performed and multiplicity adjusted $P$ values (Dunn's test) are presented. All statistical tests were two-tailed to a significance level of 5%. For detailed description of exploratory analyses, see Supplementary Table 8. For analysis of selected secondary variables, a molecular dataset was defined. The molecular dataset included all patients whose astrocytomas could retrospectively be defined molecularly according to copy number variation load (CNV-L), methylation class, and *CDKN2A/B* status.

## Reporting summary

Further information on research design is available in the Nature Research Reporting Summary linked to this paper.

## Data availability

Single-cell RNA-seq data that are associated with Fig. 4 and Extended Data Figs. 11, 12 have been deposited in the NCBI Sequence Read Archive with the accession codes SRR12880623 and SRR12880624. TCRB sequencing data that are associated with Extended Data Fig. 10 are available at https://clients.adaptivebiotech.com/pub/platten-2021-nature.

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

**Acknowledgements** We are indebted to all patients and their relatives, and all trial sites. The NOA16 trial was funded by the German Ministry of Education and Science and the National Center for Tumor Diseases (ClinicalTrials.gov number NCT02454634). We acknowledge the support of the DKFZ Light Microscopy Facility and the DKFZ Genomics and Proteomics Core Facility. We thank S. Bauer, M. Bucur, E. Hallauer, G. Haltenhof, K. Jähne, A. Siebenmorgen, S. Jünger, S. Sachse, and L. Umansky for technical support; N. Kehl for visualization of longitudinal peripheral TCR repertoires; and S. Uhlig (FlowCore Mannheim and Institute of Transfusion Medicine and Immunology) for technical FACS support. This work was supported by the German Ministry of Education and Science (National Center for Tumor Diseases Heidelberg NCT 3.0 program 'Precision immunotherapy of brain tumors' and the DKTK program), the DKFZ-MOST program (project number 2526) and the Helmholtz Program Future Topic Immunology and Inflammation (ZT-0027, WP3), the Dr. Rolf M. Schwiete Foundation and the German Research Foundation (DFG) (FOR2289: PL315/3-1), the Sonderförderlinie 'Neuroinflammation' of the Ministry of Science of Baden Württemberg, the Joint Funding Program MGH-Heidelberg Alliance in Neuro-Oncology, the Wilhelm Sander Foundation (2012.118.1), the NCT 3.0 program "Cancer immunotherapy program" program "genetically modified cells for cancer immune therapy", the Baden-Württemberg Stiftung (BWST_ISF2018-046), German Cancer Aid (70112399) to M.P., the Deutsche Forschungsgemeinschaft (DFG, German Research Foundation) – Project-ID 404521405, SFB 1389 - UNITE Glioblastoma, Work Package B01 to M.P. and T.B., the epigenetics@dkfz program, the Swiss Cancer Foundation, the Else Kröner Fresenius Foundation, the University Heidelberg Foundation, the Deutsche Forschungsgemeinschaft (DFG, German Research Foundation) – Project-ID 404521405, SFB 1389 - UNITE Glioblastoma, Work Package B03 to L.B., and German Cancer Aid (110624) to W.W. L.B. was funded by Heidelberg Medical Faculty and the Heidelberg University (Hella Bühler Award). T.B. and K.S. are supported by the Medical Faculty and University Hospital Mannheim and the University Heidelberg. K.S. was supported by a doctoral fellowship of the DKFZ. F.S. is supported by a postdoctoral fellowship of the University Hospital Heidelberg. E.G. was supported by a Marie-Curie fellowship.

**Author contributions** M.P. conceptualized and designed the trial, interpreted data, and wrote the paper. L.B. designed and performed translational analyses, analysed and interpreted data, and wrote the paper. A.W., O.S., J.H., M.M., D.K., G.T. and J.P.S. treated patients and interpreted data. T.B. performed and analysed translational analyses. L.L.C., A.F. and L.-M.R. performed primary endpoint statistical analyses. I.H., M.O.B. and M.B. performed disease assessment and analysed PsPD. F.S. and A.v.D. performed and interpreted molecular screening of tumour material and 850k arrays. K.S. performed single-cell sequencing and TCR testing, and designed the NFAT reporter assay. C.L.T. and E.G. analysed single-cell sequencing data. I.P. performed and analysed immune monitoring of primary immunogenicity endpoints and translational analyses. E.G. designed and cloned TCR vectors. S.J. performed in vitro HLA affinity analyses. G.A.B. performed HLA typing. A.S. and M.S. provided the GMP facility for generating peptide emulsion. S.S. synthesized peptides. W.W. conceptualized the trial, interpreted data, and wrote the paper.

**Funding** Open access funding provided by Deutsches Krebsforschungszentrum (DKFZ).

**Competing interests** M.P., T.B. and W.W. are inventors and patent-holders on 'Peptides for use in treating or diagnosing IDH1R132H positive cancers' (EP2800580B1).

**Additional information**
**Correspondence and requests for materials** should be addressed to M.P.

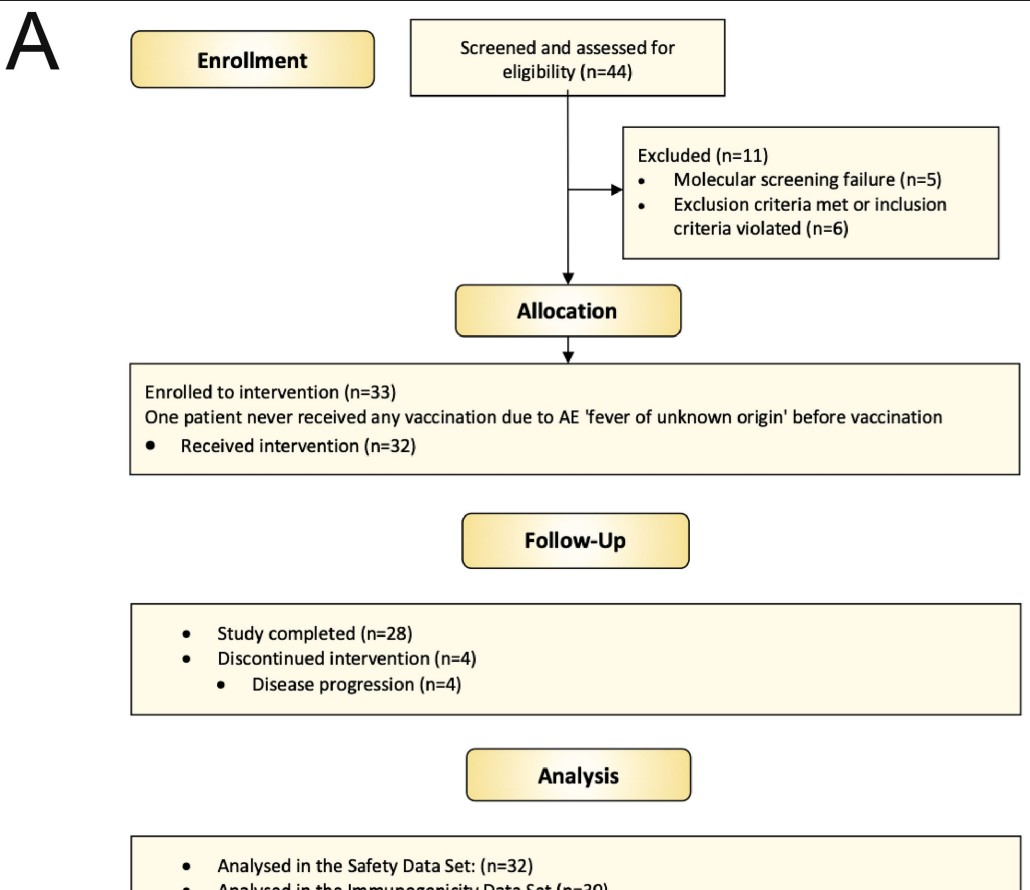

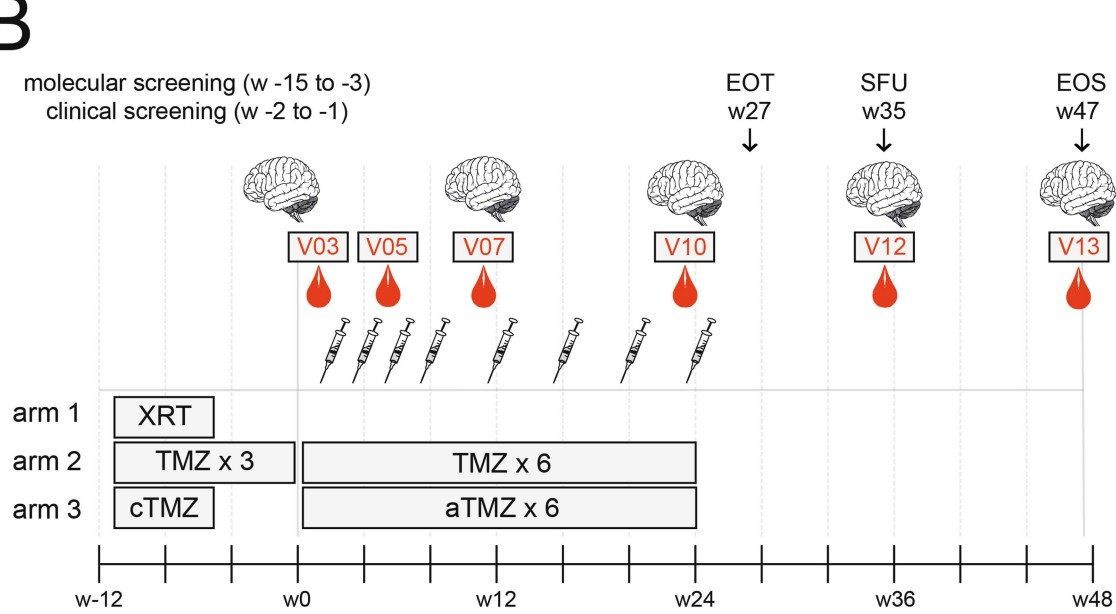

**Extended Data Fig. 1** | See next page for caption.

**Extended Data Fig. 1 | Trial design and recruitment. a**, Patient disposition CONSORT flow diagram. Forty-four patients were enrolled and screened across 7 trial sites, of which 11 were excluded. Of 33 allocated patients, 32 received the intervention. Twenty-eight patients completed the study, because four discontinued the intervention owing to disease progression. The safety dataset (SDS) for analysis contained all patients who received the intervention ($n = 32$); 30 of these were evaluable for immunogenicity and comprised the immunogenicity dataset (IDS) ($n = 30$). **b**, Study flow chart. The trial population comprised three treatment groups (arms) according to the standard therapy received. IDH1-vac was administered at V03 (week 1), V04 (week 3), V05 (week 5) V06 (week 7), V07 (week 11), V08 (week 15), V09 (week 19), and V10 (week 23). Blood for primary endpoint immunogenicity testing was drawn at V03 (baseline), V05, V07, V10, V12 (week 35, safety follow-up (SFU)), and V13 (week 47, EOS). MRI scans (represented by brain images) were performed at clinical screening, V07, V10, V12, and V13. XRT, radiotherapy; TMZ, temozolomide; cTMZ, concomitant TMZ, RT + TMZ; aTMZ, adjuvant TMZ. TMZ cycle numbers indicated.

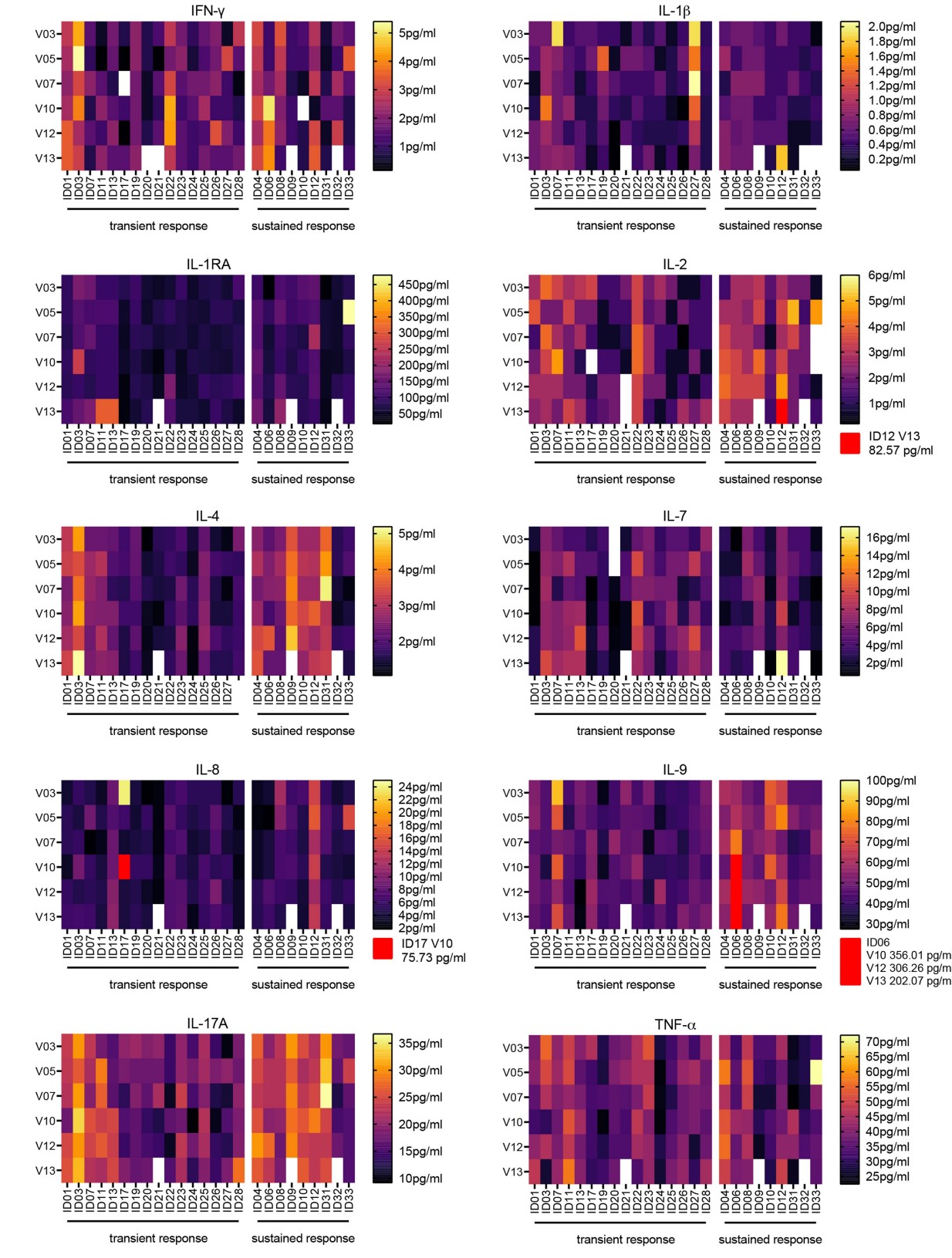

**Extended Data Fig. 2 | Serum cytokine levels during treatment with IDH1-vac.** Heat maps depicting longitudinal (V03–V13) cytokine concentrations in sera from transient (*n* = 16) and sustained (*n* = 9) T cell responder patients, measured by multiplex bead technology. For definition of transient and sustained responses, see Methods. White, sample not available; red, concentration out of depicted range.

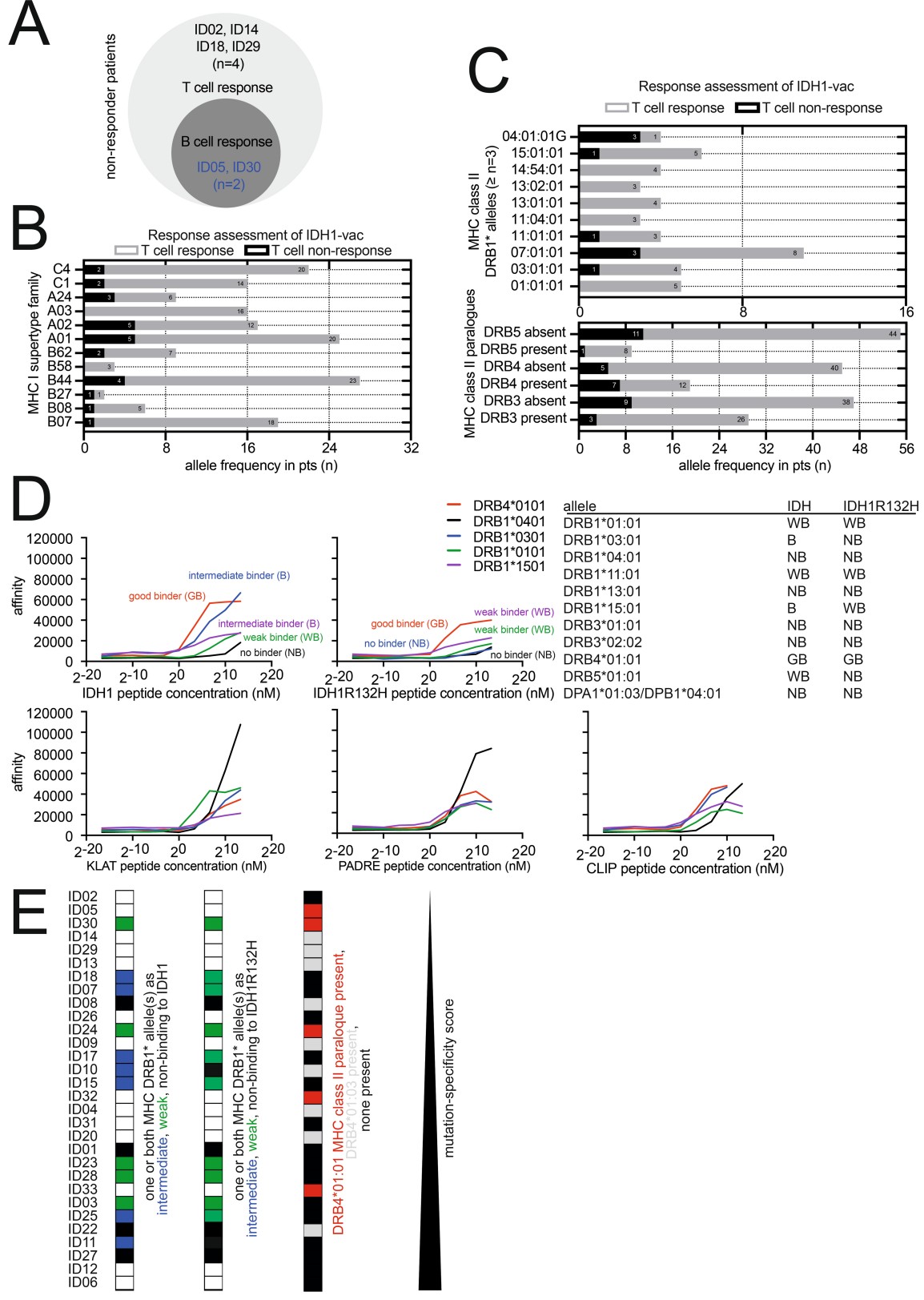

**Extended Data Fig. 3 | Relationship between MHC alleles and T cell response. a**, Venn diagram of T cell non-responders and B cell non-responders in the IDS. **b, c**, Allele prevalence of MHC class I supertype families (**b**), and MHC class II DRB1* alleles with a total prevalence of three or more, and paralogues (**c**). Grey, numbers of alleles present or absent (for paralogues) in patients with T cell responses to IDH1-vac (T cell response); black, numbers of alleles present or absent (for paralogues) in patients without T cell responses to IDH1-vac (T cell non-response). *n* (total alleles) = 64 for 32 patients in the SDS. **d**, IDH1

and IDH1(R132H) 20-mer p123–142 affinities to six MHCII DRB1* alleles and DRB3*, DRB4*, and DRB5* MHCII paralogues were assessed in vitro. Four alleles and DRB4* paralogue are shown as examples in the graphs. CLIP, KLAT, and PADRE represent positive control peptides. GB, good binder; B, intermediate binder; WB, weak binder; NB, non-binder. *n* = 1 of 2 independent experiments. **e**, Correlation analysis of in vitro MHC class II affinities with MSS. White, no affinity-tested alleles present; blue, intermediate binder(s) present; green, weak binder(s) present; black, non-binders present. *n* = 30 patients in the IDS.

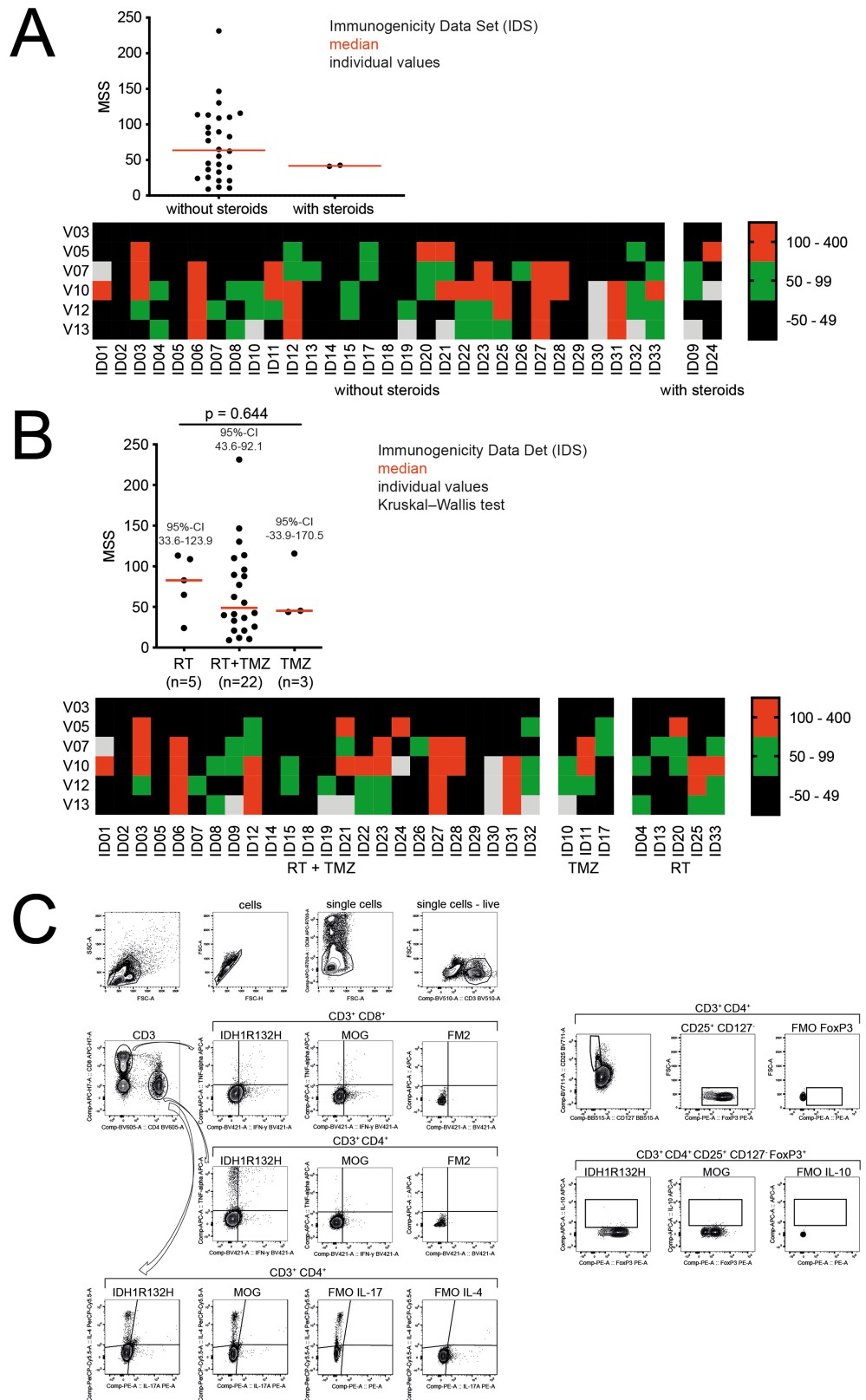

**Extended Data Fig. 4 | T cell immunogenicity and standard treatment in the IDS. a**, **b**, T cell immune responses assessed by MSS (top) and by immune response criteria for positivity over time (bottom) according to concomitant use of steroids until EOT (**a**) and primary SOC treatment (**b**). Top, individual values and median. $n$(steroids) = 2; $n$(no steroids) = 28. $n$(RT) = 5; $n$(RT + TMZ) = 22; $n$(TMZ) = 3. Two-sided Kruskal–Wallis test with Dunn's multiple comparison (**b**). **c**, Gating strategy for flow cytometric effector sub-phenotyping of peripheral IDH1-vac-induced T cells shown in Fig. 2c. FM2, fluorescence minus two; FMO, fluorescence minus one; MOG, negative control.

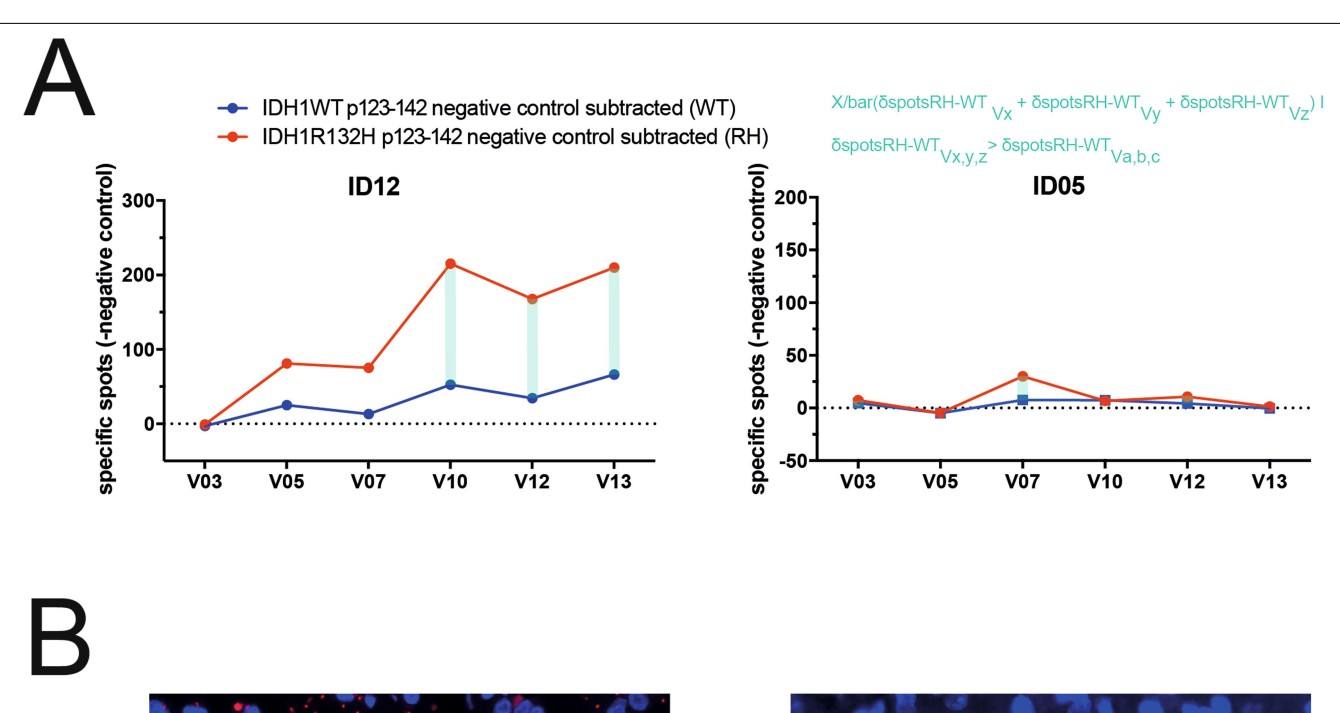

**A**

● IDH1WT p123-142 negative control subtracted (WT)
● IDH1R132H p123-142 negative control subtracted (RH)

$\bar{X}/(\delta spotsRH\text{-}WT_{Vx} + \delta spotsRH\text{-}WT_{Vy} + \delta spotsRH\text{-}WT_{Vz})$ I

$\delta spotsRH\text{-}WT_{Vx,y,z} > \delta spotsRH\text{-}WT_{Va,b,c}$

**B**

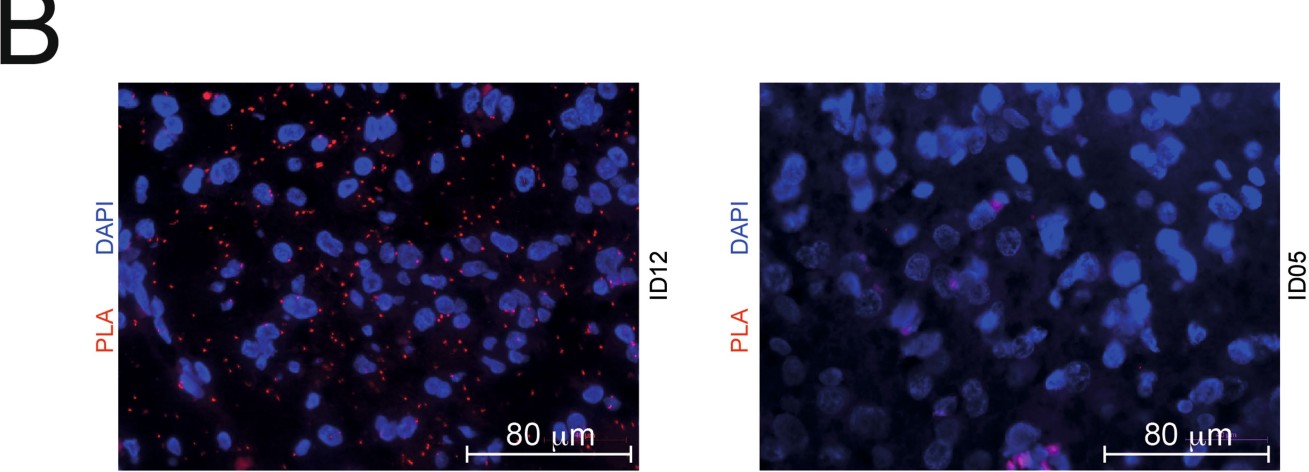

**C**

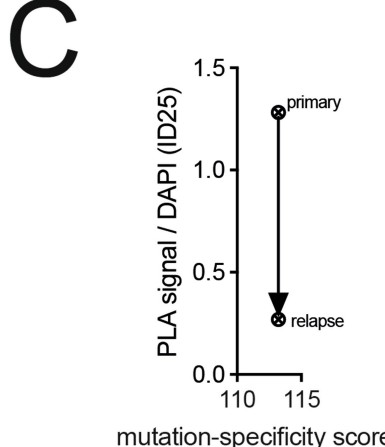

**Extended Data Fig. 5 | Definition of the MSS and PLA. a**, Exemplary MSS of patients ID12 and ID05. Mathematical differences between specific spot counts for IDH1(R132H) (red) and wild-type IDH1 (blue) are calculated (light blue) for all immunogenicity testing visits. MSS is defined as the sum of the three largest differences. For MSS of the IDS please refer to Supplementary Table 7. **b**, Exemplary PLA of patients ID12 and ID05. For relative PLA values of the IDS please refer to Fig. 2. Scale bars, 80 μm. PLA was performed once per patient. **c**, Relative PLA signals of primary and recurrent tissues from patient ID25.

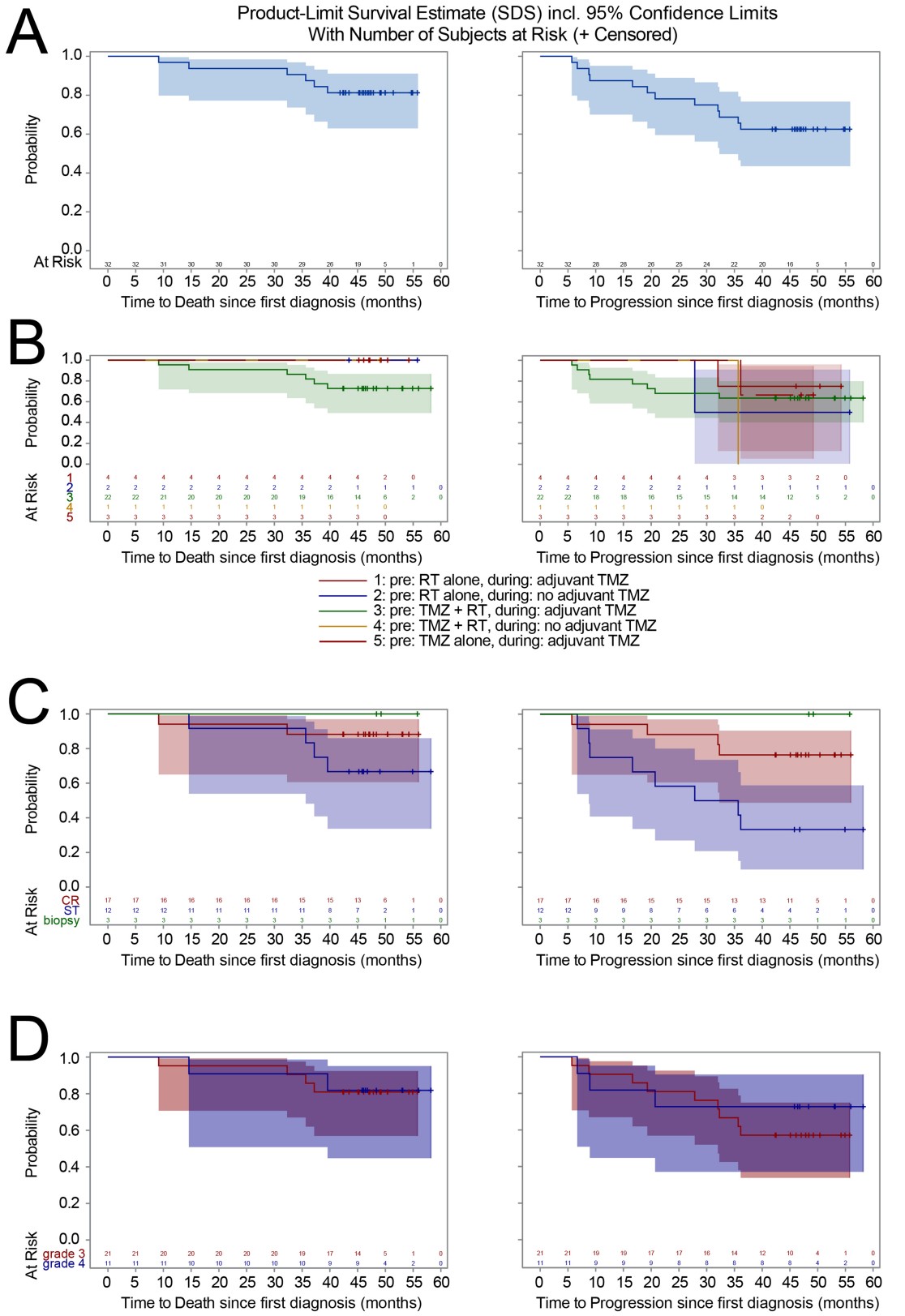

**Extended Data Fig. 6 | Probabilities of progression and death in the SDS.** Overall (left) and progression-free (right) survival estimates with number of patients at risk are shown for all patients of the SDS (**a**), according to SOC treatment (**b**), extent of resection (**c**), and WHO grade (**d**). CR, complete resection; ST, subtotal resection. $n$(all patients) = 32; $n$(RT, aTMZ) = 4; $n$(RT) = 2; $n$(TMZ + RT, aTMZ) = 22; $n$(TMZ + RT) = 1; $n$(TMZ) = 3; $n$(CR) = 17; $n$(ST) = 12; $n$(biopsy) = 3; $n$(grade 3) = 21; $n$(grade 4) = 11. Time, months from first diagnosis.

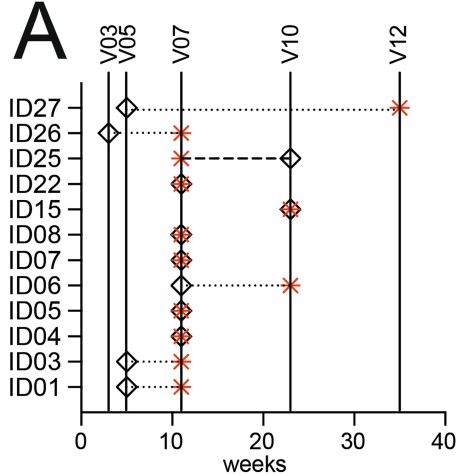

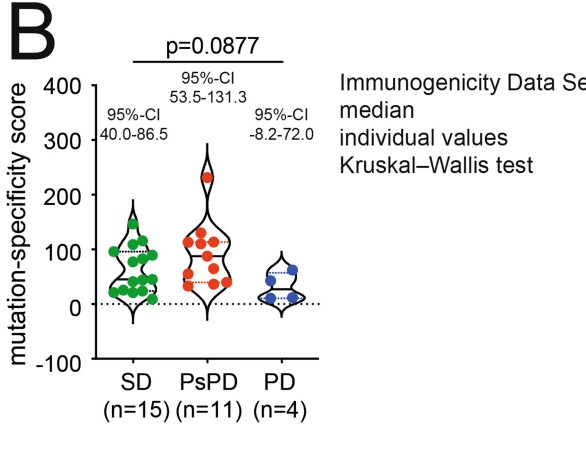

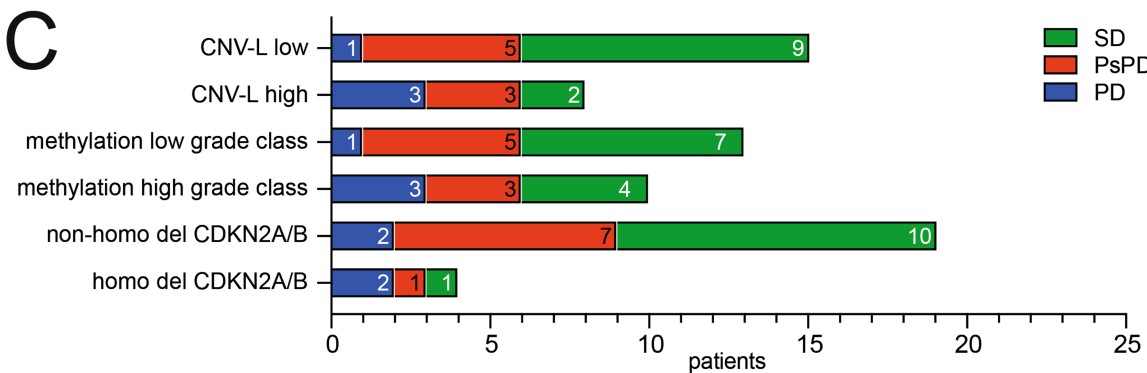

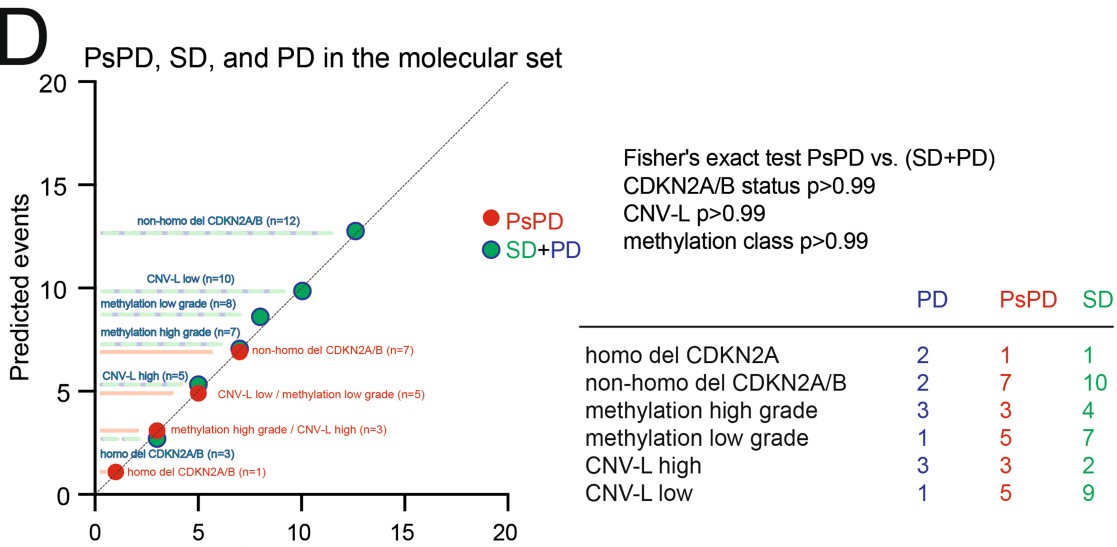

**Extended Data Fig. 7 | Determinants of pseudoprogression. a**, Time points of PsPD diagnosed by MRI and onset of early immune response are shown in weeks and study visits. Black diamond, onset of immune response; red asterisk, PsPD. Visits with immune monitoring (V03, V05, V07, V10, and V12) are depicted. MRI was performed at clinical screening, V07, V10, V12 and V13. The onset of an early immune response was defined as antibody titre ≥ 1:333 (B cell), and/or T cell response detectable after negative control subtraction. $n$ = 12 patients with PsPD. **b**, Violin plots showing MSS according to disease progression. Solid line, median; dotted lines, quartiles. $n$(SD) = 15; $n$(PsPD) = 11; $n$(PD) = 4. Two-sided Kruskal–Wallis test with Dunn's multiple comparison. **c**, Incidence of disease progression in patients with molecularly defined astrocytomas according to CNV-L, methylation class, and *CDKN2A/B* status. **d**, Correlation between predicted and actual occurrence of molecular markers in groups of disease outcomes. Two-sided Fisher's exact test (PsPD versus SD + PD). $n$(all patients) = 23 (**c**, **d**).

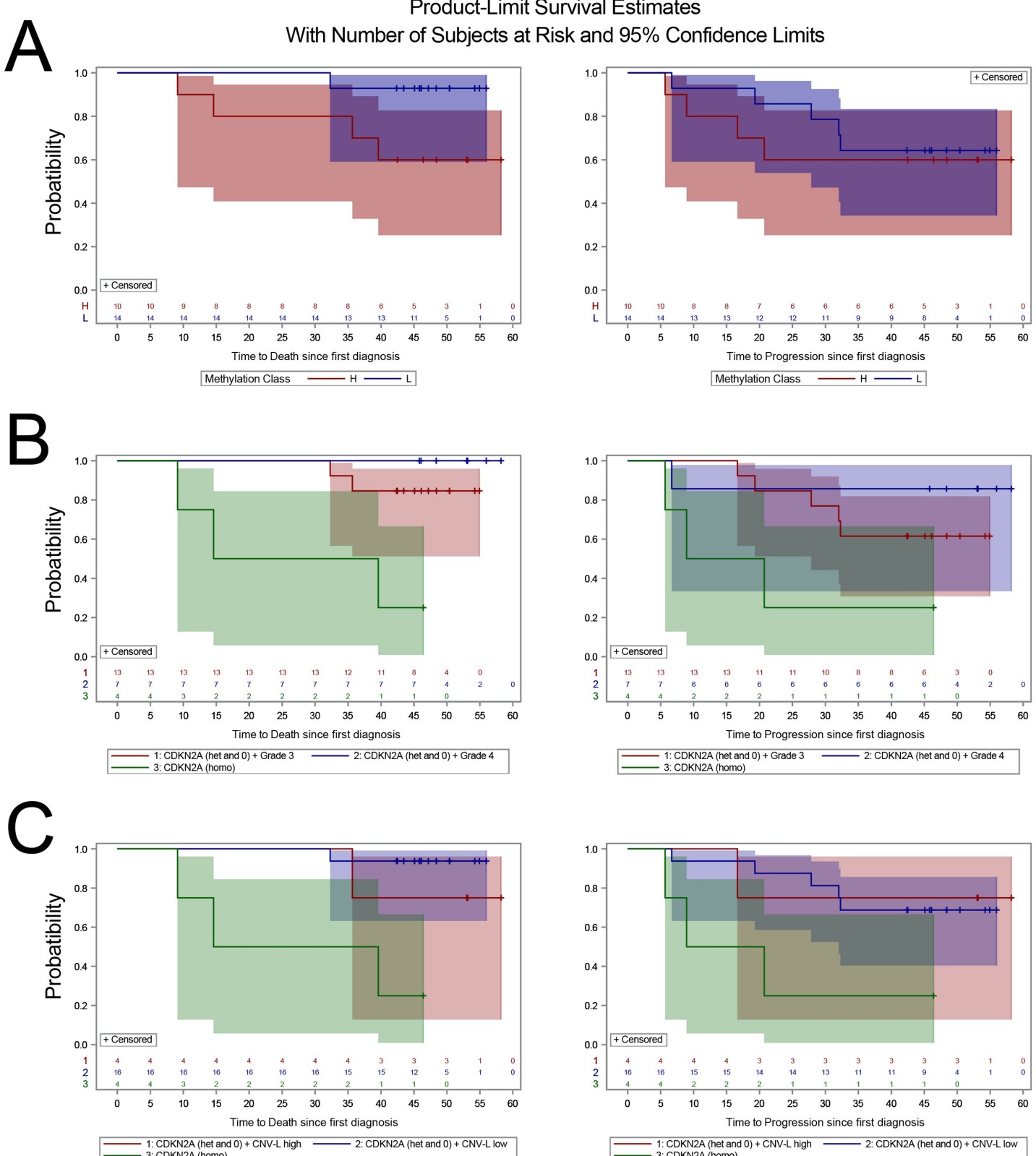

**Extended Data Fig. 8 | Probabilities of progression and death in the molecular dataset.** Overall (left) and progression-free (right) survival estimates with number of patients at risk are shown according to methylation class (**a**), *CDKN2A/B* status and grade (**b**), and *CDKN2A/B* status and CNV-L (**c**). H, methylation class high; L, methylation class low; het, heterozygous *CDKN2A/B*

deletion; 0, no *CDKN2A/B* deletion; homo, homozygous *CDKN2A/B* deletion. *n*(all patients) = 24; *n*(methylation class H) = 10; *n*(methylation class L) = 14; *n*(*CDKN2A/B* het and 0 + grade 3) = 13; *n*(*CDKN2A/B* het and 0 + grade 4) = 7; *n*(*CDKN2A/B* het and 0 + CNV-L high) = 4; *n*(*CDKN2A/B* het and 0 + CNV-L low) = 16; *n*(*CDKN2A/B* homo) = 4.

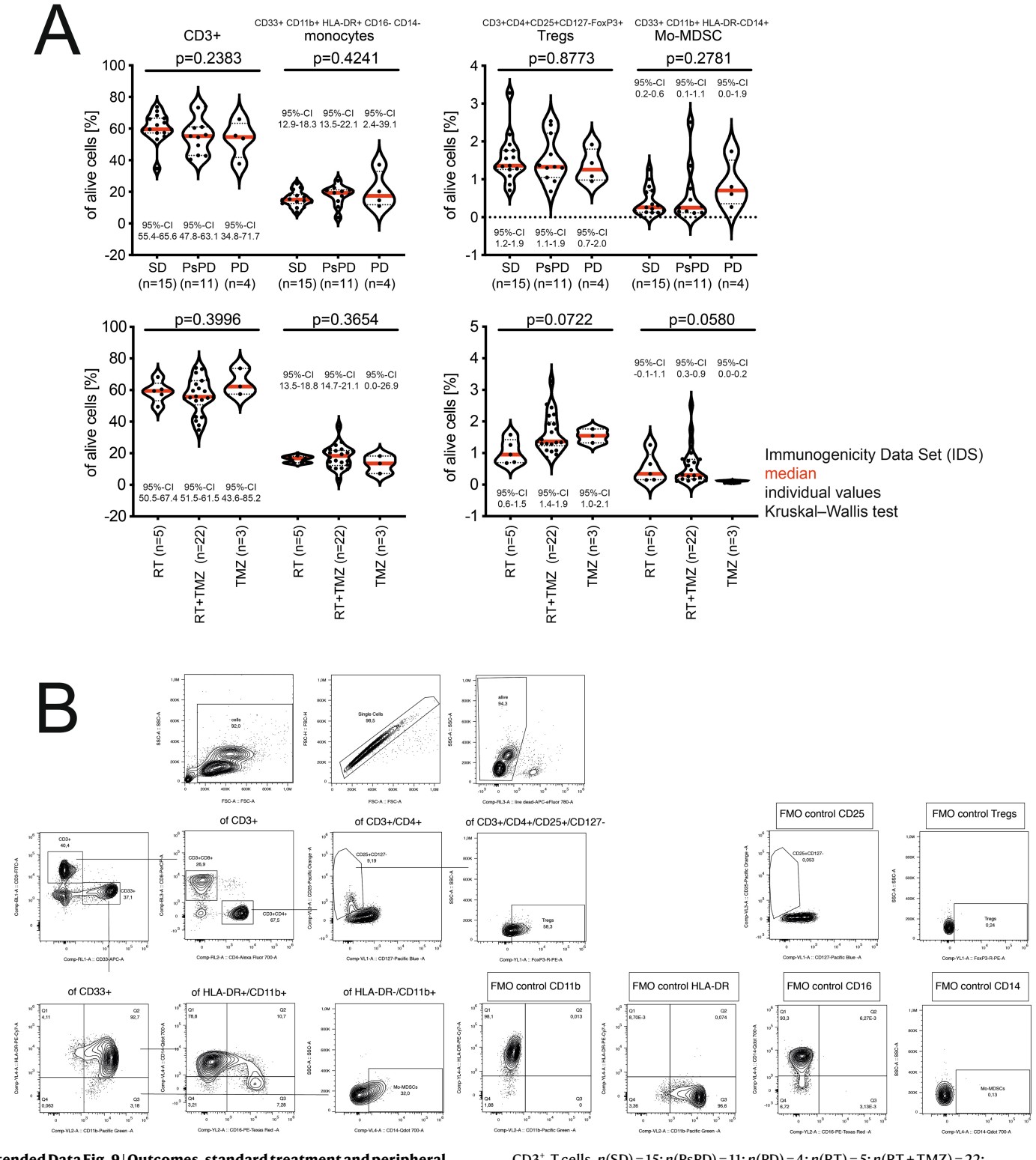

**Extended Data Fig. 9 | Outcomes, standard treatment and peripheral immune cell composition in the IDS. a**, Frequencies of T cells, monocytes, regulatory T cells (Tregs), and monocytic myeloid-derived suppressor cells (mo-MDSC) within PBMCs were determined at V07 by flow cytometry and are shown according to disease course at EOS (top) and SOC treatment (bottom).

CD3[+], T cells. *n*(SD) = 15; *n*(PsPD) = 11; *n*(PD) = 4; *n*(RT) = 5; *n*(RT + TMZ) = 22; *n*(TMZ) = 3. Red line, median; dotted lines, quartiles. Two-sided Kruskal–Wallis test with Dunn's multiple comparison. **b**, Gating strategy for flow cytometric analysis.

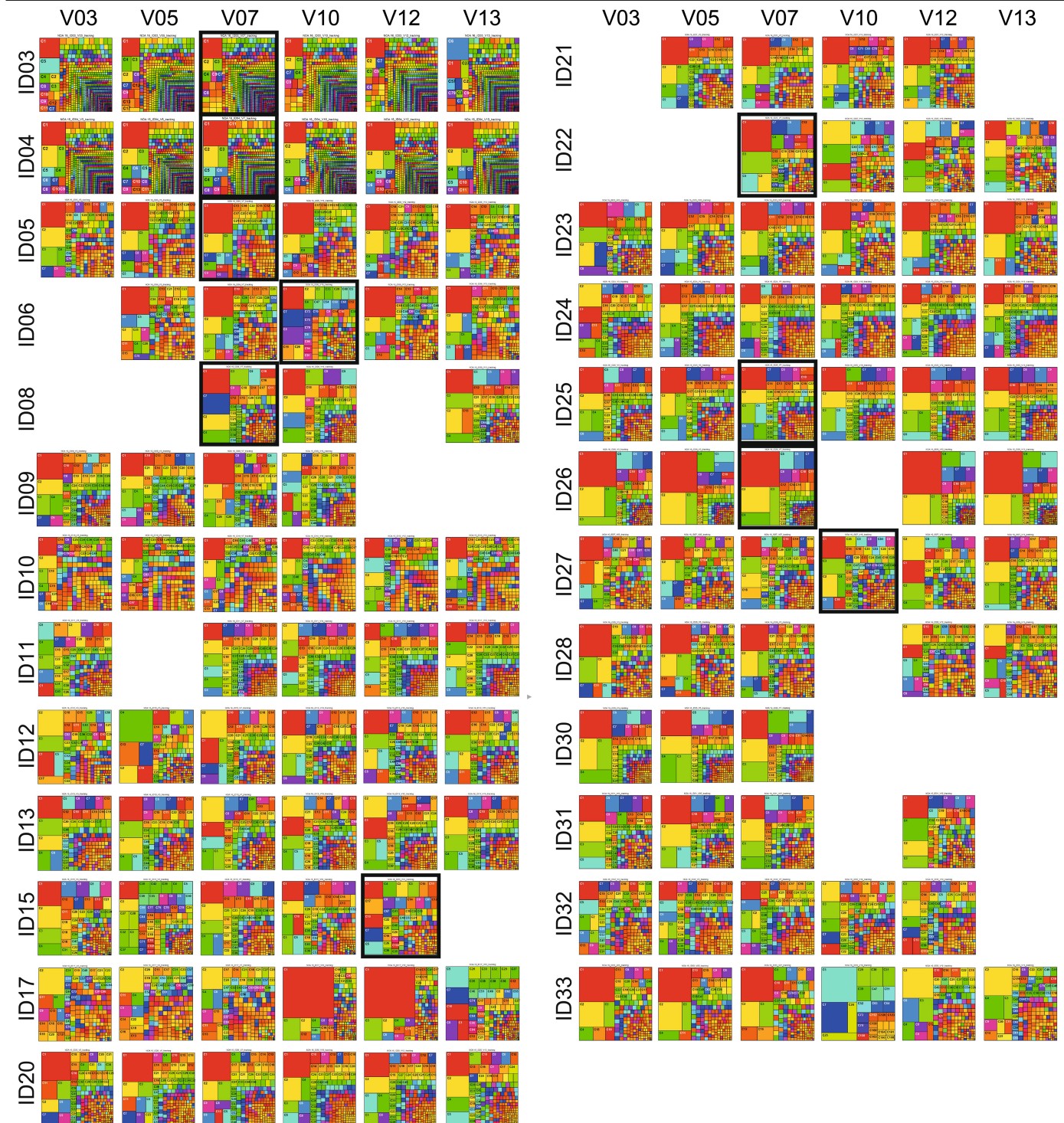

**Extended Data Fig. 10 | Longitudinal TCRB deep sequencing of PBMC.** Tree maps of longitudinal TCRB deep sequencing of PBMCs. Black outlines highlight time points of PsPD. TCRB deep sequencing was performed if PBMCs were available for exploratory analyses. *n* = 25 patients of the SDS.

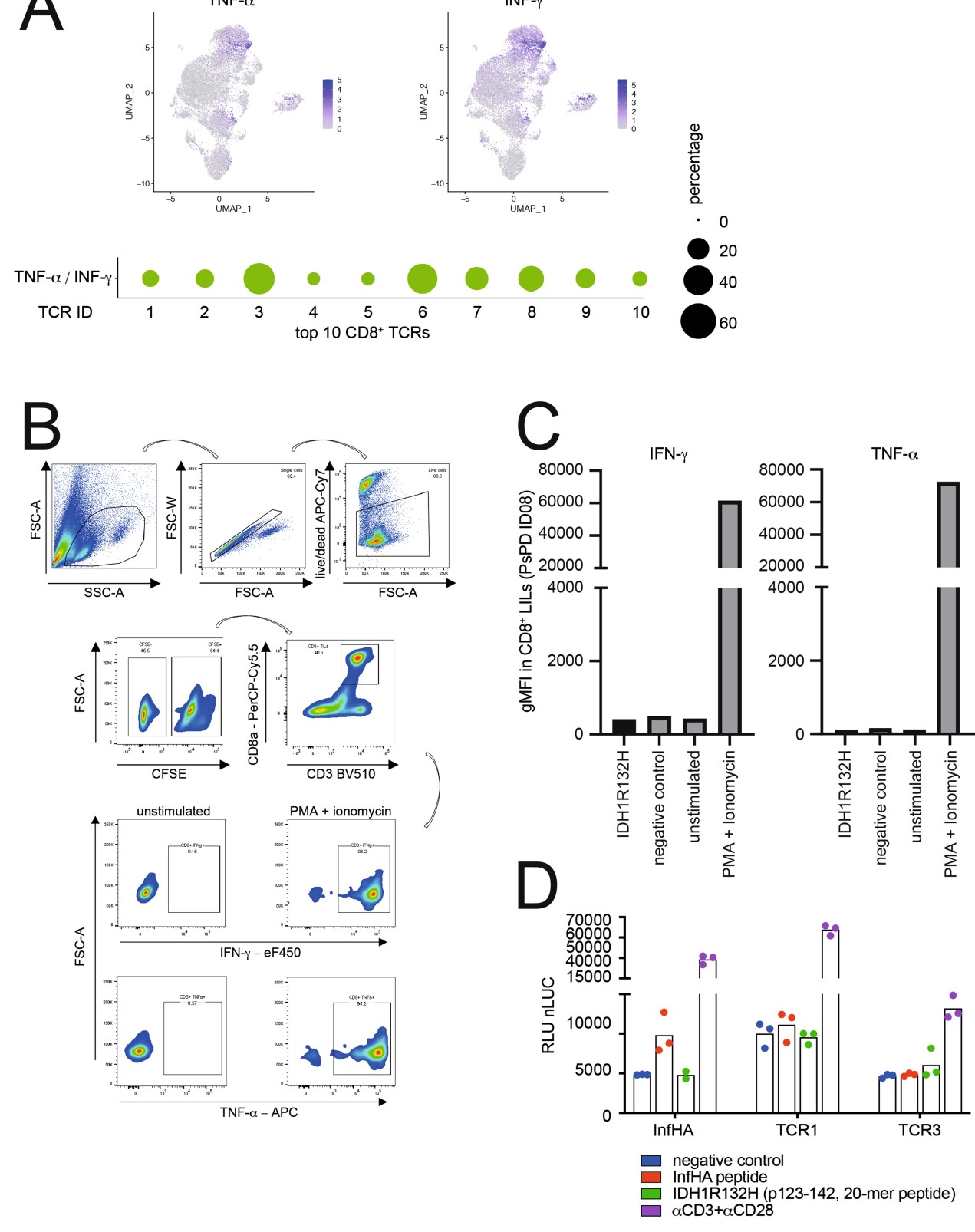

**Extended Data Fig. 11 | Ex vivo cytotoxic T cell responses, CD8⁺ T cell clone-specific cytokine production and TCR-transgenic cell reactivity of PsPD LILs from patient ID08. a**, Top, visualization of TNF- and IFNγ-expressing T cells by UMAP (Fig. 4). Bottom, relative percentages of cells expressing IFNγ and TNF among CD8⁺ cells, each expressing one of the top ten TCRs. **b**, **c**, Flow cytometric gating strategy (**b**) and quantification (**c**) of ex vivo cytotoxic T cell responses of PsPD CD8⁺ LILs from patient ID08 upon re-stimulation with IDH1(R132H). **d**, CD8⁺ T cell clonotype-retrieved TCR-transgenic cell reactivity in luciferase NFAT reporter assays. TCR1, overall top abundant ID08 PsPD CD8⁺ T cell clonotype; TCR3, ID08 PsPD CD8⁺ T cell clonotype with top TNF/IFNγ percentage (**a**). InfHA (influenza HA peptide; PKYVKQNTLKLAT) and its respective TCR-transgenic cells were used as positive control. Technical triplicates with mean, experiment performed once.

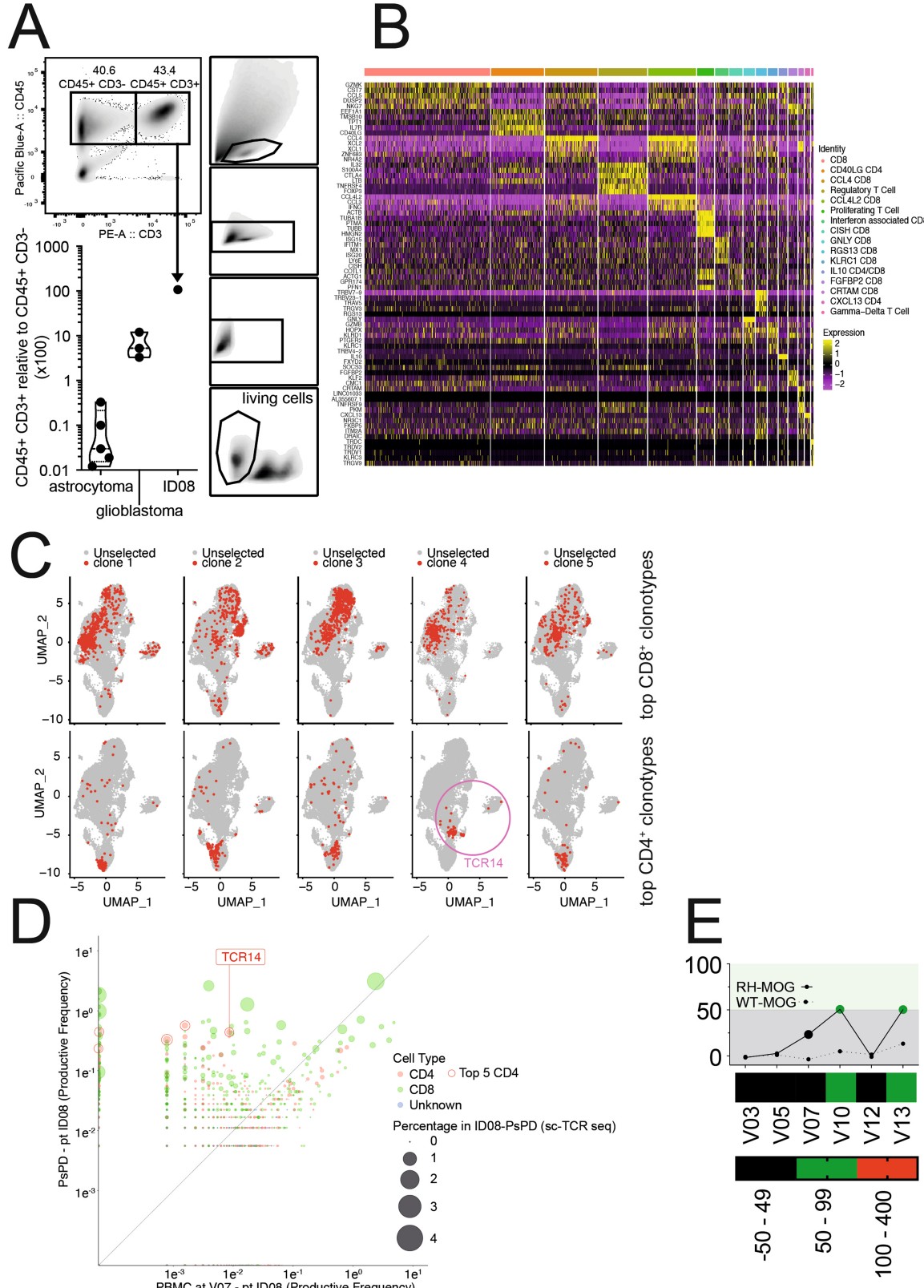

**Extended Data Fig. 12 | Single-cell RNA-TCR-seq and TCRB deep sequencing of PsPD from patient ID08. a**, Bottom left, abundance of T cells relative to non-T cells in PsPD from patient ID08 versus control IDH1(R132H)[+] astrocytoma and glioblastoma tissues. Dashed line, median; dotted lines, quartiles. Top and right, gating strategy of ID08-PsPD T cell sorting. **b**, Heat map of RNA expression of single-cell cluster-defining genes (see Fig. 4). **c**, Co-visualization of top 5 CD8[+] and CD4[+] TCR clonotypes and corresponding RNA single-T cell clusters by UMAP (see Fig. 4). **d**, Combined visualization of TCRB deep sequencing and scRNA-TCR-seq of PBMCs from patient ID08 at V07 versus LILs from PsPD of patient ID08. The frequency of TCR14 is highlighted. **e**, Longitudinal T cell response of patient ID08 assessed by IFNγ ELISpot.

# nature research

# Reporting Summary

Nature Research wishes to improve the reproducibility of the work that we publish. This form provides structure for consistency and transparency in reporting. For further information on Nature Research policies, see our Editorial Policies and the Editorial Policy Checklist.

## Statistics

For all statistical analyses, confirm that the following items are present in the figure legend, table legend, main text, or Methods section.

| n/a | Confirmed | |
|---|---|---|
| ☐ | ☒ | The exact sample size (*n*) for each experimental group/condition, given as a discrete number and unit of measurement |
| ☐ | ☒ | A statement on whether measurements were taken from distinct samples or whether the same sample was measured repeatedly |
| ☐ | ☒ | The statistical test(s) used AND whether they are one- or two-sided<br>*Only common tests should be described solely by name; describe more complex techniques in the Methods section.* |
| ☐ | ☒ | A description of all covariates tested |
| ☐ | ☒ | A description of any assumptions or corrections, such as tests of normality and adjustment for multiple comparisons |
| ☐ | ☒ | A full description of the statistical parameters including central tendency (e.g. means) or other basic estimates (e.g. regression coefficient) AND variation (e.g. standard deviation) or associated estimates of uncertainty (e.g. confidence intervals) |
| ☐ | ☒ | For null hypothesis testing, the test statistic (e.g. *F*, *t*, *r*) with confidence intervals, effect sizes, degrees of freedom and *P* value noted<br>*Give P values as exact values whenever suitable.* |
| ☒ | ☐ | For Bayesian analysis, information on the choice of priors and Markov chain Monte Carlo settings |
| ☐ | ☒ | For hierarchical and complex designs, identification of the appropriate level for tests and full reporting of outcomes |
| ☐ | ☒ | Estimates of effect sizes (e.g. Cohen's *d*, Pearson's *r*), indicating how they were calculated |

*Our web collection on statistics for biologists contains articles on many of the points above.*

## Software and code

Policy information about availability of computer code

| | |
|---|---|
| Data collection | single cell seq: Chromium Single Cell V(D)J Reagent kit v1 chemistry (10x Genomics), sequenced on HiSeq2500 rapid and HiSeq4000 platforms (Illumina).<br>TCRB seq: hsTCRB Kit (Adaptive Biotechnologies), sequenced on Illumina MiSeq.<br>Flow Cytometry: BD FASCDiva version 8.0 (FACSAria IIu for sorting) or version 9.0 (FACSCanto II for CD8+ LIL analysis), Attune Nxt software version 2.7 (Attune Nxt for peripheral immune monitoring), and BD FACSuite version 1.3 (Lyric for recall assay) |
| Data analysis | single cell seq: processed with cellranger pipeline (version 3.1.0) with GRCh38 genome assembly (version 3.0.0, 10x Genomics). analysis of filtered matrices with Seurat.transformation and normalization with sctransform (Seurat version 3.2.0). Gene expression analysis of clusters with MAST (https://github.com/RGLab/MAST) using R version 3.6.3.<br>TCRB seq: Data processing (demultiplexing, trimming, gene mapping) was done using Adaptive Biotechnologies´ proprietary platform. Data was visualized using the Treemap Visualization package version 2.4.2 (https://cran.r-project.org/web/packages/treemap/index.html).<br>Flow cytometry: FlowJo V.10.5.0.<br>exploratory statistical analysis: GraphPadPrism V.9.0.0. |

For manuscripts utilizing custom algorithms or software that are central to the research but not yet described in published literature, software must be made available to editors and reviewers. We strongly encourage code deposition in a community repository (e.g. GitHub). See the Nature Research guidelines for submitting code & software for further information.

## Data

Policy information about availability of data

All manuscripts must include a data availability statement. This statement should provide the following information, where applicable:

- Accession codes, unique identifiers, or web links for publicly available datasets
- A list of figures that have associated raw data
- A description of any restrictions on data availability

single cell RNA and VDJ sequencing data that are associated with Figure 4 and ED Figures 11, 12 have been deposited in the NCBI Sequence Read Archive with the accession codes SRR12880623 and SRR12880624, and are publicly accessible. TCRB sequencing data that are associated with ED Figures 10, 12 is available at https://clients.adaptivebiotech.com/immuneaccess

# Field-specific reporting

Please select the one below that is the best fit for your research. If you are not sure, read the appropriate sections before making your selection.

☒ Life sciences  ☐ Behavioural & social sciences  ☐ Ecological, evolutionary & environmental sciences

For a reference copy of the document with all sections, see nature.com/documents/nr-reporting-summary-flat.pdf

# Life sciences study design

All studies must disclose on these points even when the disclosure is negative.

| | |
|---|---|
| Sample size | Sample size estimation was primarily based on the accuracy requirements for the primary endpoint immune response (responder rate) to the IDH1 peptide vaccine. sample size was adjusted for non-evaluable patients (see Data Exclusions). Estimation that 70% of patients evaluable for immunogenicity testing will be evaluable for all time points (Macdonald D, Cascino T, Schold SJ, et al: Response criteria for phase II studies of supratentorial malignant glioma. J Clin Oncol 8:1277-1280, 1990.) 21 patients sufficient for immunogenicity testing with all time points, i.e. 30 evaluable patients to be enrolled. due to expected dropout rate of 20% (due to progression or other reasons), 39 patients to be recruited. |
| Data exclusions | Of all 32 patients treated (= safety data set), 2 patients were excluded from immunogenicity testing, because they were not evaluable, because not enough time points were eligible to immunogenicity testing. A patient was pre-defined to be evaluable if study completed until and incl. visit 7, received at least 4 vaccinations and all blood samples collected for immunogenicity testing, OR received 6 vaccinations and baseline plus 2 blood samples collected for immunogenicity testing. |
| Replication | Findings concerning patient outcomes and immunogenicity at defined time points, and all related analyses using patient blood samples or derivatives thereof cannot be reproduced due to sample limitations. TCR testing using TCR14 expressing Jurkat T cell line was reproduced three times with similar outcome. |
| Randomization | Phase 1 study, i.e. All patients received IDH1 vaccination. In addition, they received standard of care treatment prior to enrollment as decided by the local investigator and the patient. 3 types of standard of care treatment resulted in 3 treatment groups, all receiving the exactly same trial related intervention.<br>for PLA, tissue slides of primary tumors were selected based on availablity.<br>for TCRB deep seq, PBMC samples were selected based on availablity.<br>for recall assay, samples were selected based on availability and MSS (all available samples within top 10 highest MSS) and cells from 2 aliquots of the same sample were pooled before allocation to stimulus treatment and staining.<br>for recall assay, ID08 LIL ELISpot, ID08 CD8+ LIL FACS, and TCR14 NFAT reporter assay, cells were allocated to stimulus treatment randomly.<br>for all flow cytometry (incl. FACS), cells were allocated to different stainings (full stain panels, staining controls) randomly. |
| Blinding | single arm, open label trial, i.e. neither patients nor clinical nor immunogenicity investigators were blinded concerning IDH1 vaccination (=trial related intervention). With respect to standard of care treatment groups, immunogenicity investigators were blinded. All primary endpoint analyses were done in a blinded fashion. Exploratory analyses e.g. immunological phenotping, were performed non-blinded with respect to immune response detectable in the sample, because samples for these analyses were selected based on immune response. |

# Reporting for specific materials, systems and methods

We require information from authors about some types of materials, experimental systems and methods used in many studies. Here, indicate whether each material, system or method listed is relevant to your study. If you are not sure if a list item applies to your research, read the appropriate section before selecting a response.

## Materials & experimental systems

| n/a | Involved in the study |
|---|---|
| ☐ | ☒ Antibodies |
| ☐ | ☒ Eukaryotic cell lines |
| ☒ | ☐ Palaeontology and archaeology |
| ☒ | ☐ Animals and other organisms |
| ☐ | ☒ Human research participants |
| ☐ | ☒ Clinical data |
| ☒ | ☐ Dual use research of concern |

## Methods

| n/a | Involved in the study |
|---|---|
| ☒ | ☐ ChIP-seq |
| ☐ | ☒ Flow cytometry |
| ☐ | ☒ MRI-based neuroimaging |

# Antibodies

| Antibodies used | Flow cytometry antibodies:<br>anti-CD3-FITC (clone UCHT1, cat # 300452), anti-CD4-Alexa Fluor700 (clone RPA-T4, cat # 300526), anti-CD8-PerCP (clone RPA-T8, cat # 301030), anti-CD11b-BV510 (clone M1/70, cat # 101263), anti-HLA-DR-PE-Cy7 (clone L243, cat # 307616), anti-CD14-BV711 (clone M5E2, cat # 301838), anti-CD16-PE/Dazzle594 (clone 3G8, cat # 302054), anti-CD25-BV605 (clone BC96, cat # 302632), anti-CD33-APC (clone P67.6, cat # 366606), anti-CD127-BV421 (clone A019D5, cat # 351310), anti-Foxp3-PE (clone 206D, cat # 320108) (all BioLegend), anti-CD45-eFluor450 (clone 2D1, cat # 48-9459-42, eBioscience), anti-CD3-PE (clone HIT3a, cat # 300308, BioLegend), anti-CD3-BV510 (clone HIT3a, cat # 564713), anti-CD4-BV605 (clone SK3, cat # 566908), anti-CD8-APC-H7 (clone SK1, cat # 560179), anti-IFNgamma-BV421 (clone 4S.B3, cat # 564791), all BD Biosciences, anti-TNFalpha-APC (clone MAb11, cat # 502912, Biolegend), anti-IL17-PE (clone N49-653, cat # 560486), anti-IL4-PerCP-Cy5.5 (clone 8D4-8, cat # 561234), anti-CD25-BV711 (clone 2A3, cat # 563159), anti-CD127-FITC (clone HIL-7R-M21, cat # 560549), anti-FoxP3-PE (clone 259D/C7, cat # 560046), anti-IL10-APC (clone JES3-19F1, cat # 554707), anti-CD3-BV510 (clone HIT3a, cat # 564713) (all BD Biosciences), anti-CD8-PerCP-Cy5.5 (clone RPA-T8, cat # 45-0088-42), anti-TNFalpha-APC (clone MAb11, cat # 17-7349-82), and anti-IFNgamma-eFluor450 (clone 4S.B3, cat # 48-7319-42) (all ebioscience).<br>PLA antibodies:<br>anti-IDH1R132H (clone H09, cat # DIA-H09, Dianova), anti-HLA-DR (clone EPR3692, cat # ab92511, Abcam).<br>ELISA antibodies:<br>mouse anti-IDH1R132H (clone H09, cat # DIA-H09, Dianova), sheep anti-mouse IgG-HRP (polyclonal secondary, cat # NXA931, Amersham) and goat anti-human IgG-Fc-HRP (polyclonal secondary, cat # A80-104P, Bethyl Laboratories, Inc.). |
|---|---|
| Validation | all antibodies were titrated prior to use as advised by supplier in the material data sheet or used according to manufacturer's instructions, and, for flow cytometry, scaled up according to cell numbers.<br>Flow cytometry antibodies: each lot is quality control tested by extra- or intra-cellular flow cytometry using positive control samples and appropriate isotype control stainings.<br>anti-IDH1R132H (clone H09, cat # DIA-H09, Dianova): approved for in vitro diagnostic use with IHC in Europe.<br>anti-HLA-DR (clone EPR3692, cat # ab92511, Abcam): tested and guaranteed application for IHC on FF/PFA fixed paraffin-embedded sections and immunofluorescence.<br>secondary ELISA antibodies: Bethyl: By immunoelectrophoresis and ELISA this antibody reacts specifically with human IgG. Cross reactivity with IgM, IgA and light chains is less than 0.1%. This antibody may cross react with IgG from other species. Amersham: Horseradish peroxidase (HRP) conjugated antibodies are highly species specific antibodies optimized for use with Amersham ECL Western blotting detection reagents. |

# Eukaryotic cell lines

Policy information about cell lines

| Cell line source(s) | Jurkat delta 76, obtained from TRON gGmbH |
|---|---|
| Authentication | authenticated using the Multiplexion STR profiling and compared to normal Jurkat cells |
| Mycoplasma contamination | cells were regularly tested for mycoplasm contamination and tested negative at all time points |
| Commonly misidentified lines (See ICLAC register) | none. |

# Human research participants

Policy information about studies involving human research participants

| Population characteristics | full details for all patients incl. non-evaluable patients, are provided in Table 1 of the manuscript.<br>62.5 % male, 37.5% female. mean age was 40.4 (SD 8.95) years. 71.9 % received radiotherapy (RT) plus chemotherapy with temozolomide (TMZ), 9.4 % received TMZ alone, 18.8% received RT alone. 65.6% patients had WHO grade 3, 34.4% patients had WHO grade 4 astrocytomas. 71.9 % of tumors located at frontal lobe. 53.1 % receied complete resection, 37.5 % receivved subtotal resection, 9.4 % received biopsy. |
|---|---|
| Recruitment | patients were recruited at 8 trial centers in Germany (see ED Table 1), based on molecular and clinical inclusion criteria: presence of a histologically confirmed IDH1R132H+ glioma (with or without measurable residual tumor after resection or |

biopsy) with absence of chromosomal 1p/19q co-deletion and loss of nuclear ATRX expression in the tumor tissue (subgroup of molecular astrocytoma without positive prognostic factors). In addition, inclusion criteria were:
patients receiced standard of care treatment (RT+TMZ, TMZ alone, or RT alone) prior to enrollment; at least 18 years old; women of child-bearing potential (WOCBP) must provide a negative pregnancy test within 72 h prior to start of IDH1 vaccination; WOCBP and their partners must use birth control method (failure rate below 1% per year).
exclusion criteria included: concomitant treatment with dexamethasone (or equivalent) > 2 mg/day, Karnofsky Performance Status (KPS) < 70, progressive (incl. pseudoprogression) or recurrent disease after standard of care treatment, experimental treatment of the tumor; grade 2 or higher CTCAE v4.0 laboratory values for hematology, kiver, or renal function.
For complete list of exclusion criteria, please refer to the study protocol.
Patients agreeing to trial methods are normally well-informed and motivated to comply with study procedures, which may influence results in a way of better interpretability.

| Ethics oversight | The study was approved by the national regulatory authority (Paul-Ehrlich Institut) and the institutional review board at each study site (Ethikkommission), namely: Ethikkommission der medizinischen Fakultät Heidelberg (Heidelberg), Ethik-Kommission Albert-Ludwigs-Universität Freiburg (Freiburg), Ethik-Kommission des Landes Berlin (Berlin), Ethik-Kommission der Medizinischen Fakultät der Universität Duisburg-Essen (Essen), Ethik-Kommission der Medizinischen Fakultät "Carl Gustav Carus" (Dresden), Ethikkommission des Fachbereichs Medizin der Goethe-Universität Frankfurt am Main (Frankfurt), Ethik-Kommission an der Medizinischen Fakultät der Eberhard-Karls-Universität und am Universitätsklinikum Tübingen (Tübingen). The study was conducted in accordance with the Good Clinical Practice guidelines of the International Conference on Harmonisation. All participants provided written signed informed consent. |

Note that full information on the approval of the study protocol must also be provided in the manuscript.

# Clinical data

Policy information about clinical studies

All manuscripts should comply with the ICMJE guidelines for publication of clinical research and a completed CONSORT checklist must be included with all submissions.

| Clinical trial registration | NCT-2013-0216 |

| Study protocol | not publicly accessible. all relevant information can be found on ClinicalTrials.gov |

| Data collection | patients were recruited at 8 sites across Germany: Heidelberg (14 patients), Freiburg (2 patients), Berlin (4 patients), Essen (3 patients), Dresden (3 patients), Frankfurt (3 patients), Tuebingen (4 patients) (for details refer to ED Table 1). Screening was between July 2015 and September 2016. Enrollement was between September 2015 and October 2016. Last patient completed the study in September 2017. Follow-up is ongoing; FU clinical data of the manuscript are described as of June 2020. Imunogenicity analyses and related analyses were completed in June 2020. |

| Outcomes | primary objectives were safety and tolerability; and immunogenicity of the IDH1 vaccine.
Saftey measures were assessed by medical review at each visit (every 2 or 4 weeks for 8 visits during vaccinations; 4, 8, and 12 weeks after last vaccination), incl. reporting of adverse events (AEs), concomitant medications. All AEs graded according to National Cancer Institute Common Terminology Criteria for Adverse Events (CTCAE) version 4.0. Primary endpoint regime limiting toxicity (RLT) defined as: - any injection site reaction of CTCAE grade 4; any injection site reaction of CTCAE grade 3 that persists after two weeks; any other hypersensitivity, anaphylaxis or local allergic reaction ≥ CTCAE grade 3; brain edema (CTCAE grade 4); autoimmunity ≥ CTCAE grade 3; ≥ CTCAE grade 3 toxicity to organs other than the bone marrow, but excluding grade 3 nausea, grade 3 or 4 vomiting in patients who have not received optimal treatment with antiemetics, grade 3 or 4 diarrhea in patients who have not received optimal treatment with antidiarrheas, grade 3 fatigue; death; that is definitely/certainly, probably, or possibly related to the IDH1 vaccine. patients with RLT removed from treatment, no dose-de-escalation but skipping of vaccines allowed due to AEs.
Immunogenicity endpoint (induction of presence of IDH1-reactive T cells or binding antibodies) was assessed by ELIspot and ELISA from blood PBMC and serum collected at baseline, 2 or 4 weeks after each vaccination, and 12 and 24 weeks after last vaccination. T cell immunogenicity was defined as specific (negative control subtracted) spot forming unit SFU count of at least 50 at any time point but baseline if no spontaneous response detectable. In case of spontaneous response, SFU post-IDH1-vaccination were defined to be at least 3-fold above baseline. For antibodies, optical density of at least 5-fold above negative control at any time point was defined positive.
secondary objectives were (1) to seek evidence of immunogenicity by assessing the IDH1R132H-specific T-cell and antibody response measured by IFN-γ ELISpot and ELISA, respectively, at all time points of blood withdrawal; (2) to evaluate clinical outcome by assessing the progression-free survival (PFS) and overall response rate (ORR) according to the response evaluation criteria defined as follows: ORR, defined as the proportion of patients showing complete response (CR), partial response (PR) or stable disease (SD) at end of study (EOS) compared to the baseline value for ORR under trial drug. ORR analysis will be based on the central disease assessment according to the RANO criteria; and (3) to analyze the association between immunogenicity and the clinical outcome parameters. |

# Flow Cytometry

## Plots

Confirm that:

☒ The axis labels state the marker and fluorochrome used (e.g. CD4-FITC).

☒ The axis scales are clearly visible. Include numbers along axes only for bottom left plot of group (a 'group' is an analysis of identical markers).

☒ All plots are contour plots with outliers or pseudocolor plots.

☒ A numerical value for number of cells or percentage (with statistics) is provided.

## Methodology

| | |
|---|---|
| Sample preparation | peripheral immune monitoring of PBMC: PBMC were isolated by ficoll gradient centrifugation from heparin blood, frozen in 50% freezing medium A (60% X-Vivo 20 + 40% FCS) and 50% medium B (80% FCS + 20% DMSO) and stored in liquid nitrogen at -140°C until analysis.<br>recall assay of PBMC: PBMC were thawn, rested for 4 h in X-Vivo medium, and seeded into 96-well U-bottom plates. 2x106 PBMCs were stimulated with 2 µg peptide per well using IDH1R132H(p123-142), MOG(p35-55) as negative control, or CEFT peptide pool (concentration, jpt) as positive control for 2 h before adding 10 µg/ml brefeldin A (Sigma-Aldrich, order no. B6542) and 1x GolgiStop (BD Bioscience). Cells were incubated additional 12 h<br>Leukocytes from lesion: tissue was dissected into small pieces (2x2mm) and transferred to 24-well tissue culture treated plates – 3 pieces per well in 2 ml human TIL medium (RPMI1640 (Pan Biotec) + 10% Human serum (Sigma Aldrich) + 2mM L-glutamine +1.25 µg/ml Amphotericin B (both Gibco) + 1000U/ml IL-2 (Proleukin)) containing 30ng/ml anti-human CD3 (clone OKT-3, (eBioscience)). Medium was exchanged every 2-3 days and tissue pieces removed on day 7. LILs that migrated out of the tumor into the medium were further expanded until day 14 and cryopreserved as above. |
| Instrument | peripheral immune monitoring of PBMC: Attune Nxt (Thermo Fisher Scientific); recall assay of PBMC: Lyric (BD); sorting of T cells: FACSAria IIu (BD); CD8+ LIL testing: Canto II (BD) |
| Software | collection: Attune Nxt software version 2.7 (Thermo Fisher Scientific) (peripheral immune monitoring); BD FACSDiva software version 8.0 (sorting of T cells) or version 9.0 (CD8+ LIL testing); BD FACSuite version 1.3 (recall assay).<br>analysis: FlowJo version 10.5.0. |
| Cell population abundance | relevant post-sorted cell fractions, i.e. CD3+ T cells, were subjected to scRNAsequencing. based on RNA expression signatures, after further removal of stressed cells (by expression of heat shock proteins), 99.42% of cells were T cells. Non-T cells were excluded from analysis. |
| Gating strategy | peripheral immune monitoring of PBMC:<br>gated on lymphocytes based on size and granularity (FSS vs SSC) --> gated on single cells --> gated on live cells, i.e. dead stain negative cells --> gated on: A. CD3+ --> CD8+/CD4+ --> CD4+ gated on CD25+ CD127 --> gated on FoxP3+ cells = Treg, and B. CD33+ --> gated on a. CD11b+ HLA-DR+ --> CD14-/CD16- cells = monocytes, and b. CD11b+/HLA-DR- --> CD14+ cells = Mo-MDSC<br>recall assay:<br>gated on lymphocytes based on size and granularity (FSS vs SSC) --> gated on single cells --> gated on live cells, i.e. dead stain negative cells --> gated on CD3+ --> gated on A. CD4- CD8+ --> gated on IFNgamma+ and TNFalpha+, and B. CD4+ CD8- --> gated on a. IFNgamma+ and TNFalpha+, b. IL-4+ and IL-17+, and c. CD25+ CD127- --> gated on FoxP3+ --> gated on IL10+<br>sorting of T cells:<br>gated on lymphocytes based on size and granularity (FSS vs SSC) --> gated on single cells --> gated on live cells, i.e. dead stain negative cells --> gated on CD45 and CD3 double positive cells. live, CD45, CD3 gates based on FMO controls.<br>CD8+ LIL testing:<br>gated on lymphocytes based on size and granularity (FSS vs SSC) --> gated on single cells --> gated on live cells, i.e. dead stain negative cells --> gated on CFSE+ LIL (to distinguish from REP antigen-presenting cells) --> gated on CD3+ CD8+ --> gated on A. IFNgamma+ and B. TNFalpha+ |

☒ Tick this box to confirm that a figure exemplifying the gating strategy is provided in the Supplementary Information.

# Magnetic resonance imaging

## Experimental design

| | |
|---|---|
| Design type | n/a. Structural MRI |
| Design specifications | At least 6 MRI scans were acquired for each patient (pre study; clinical screening, visit 7,10,12,13) |
| Behavioral performance measures | n/a. MRI used solely for diagnosis of disease progression. |

## Acquisition

Imaging type(s)  
T2-w; FLAIR; contrast enhanced T1-w

Field strength  
3 Tesla

Sequence & imaging parameters  
Parameter T2-w (TSE); TE: 88; TR: 5.280; Flip angle: 180; FOV: 230 × 230; Matrix size: 256 × 173; Slice thickness: 5; No. of averages 1; In-plane resolution 0.7 Orientation Axial (2D) Duration 1:15 (min:s)

Parameter FLAIR; TE: 135; TR: 8.500; Flip angle: 170; FOV: 230 x 170; Matrix size: 320 × 216; Slice thickness: 5 No. of averages 1 In-plane resolution: 1 Orientation Axial (2D) Duration 2:52 (min:s)

T1 (mpRAGE): TE: 3.57; TR: 1.770; Flip angle: 15; FOV: 256 × 256; Matrix size: 320 × 272; Slice thickness: 1 No. of averages: 1; In-plane resolution 1; Orientation sagittal(3D); Duration 3:27 (min:s)

Area of acquisition  
whole brain scan

Diffusion MRI    ☐ Used    ☒ Not used

## Preprocessing

Preprocessing software  
n/a

Normalization  
n/a

Normalization template  
n/a

Noise and artifact removal  
n/a

Volume censoring  
n/a

## Statistical modeling & inference

Model type and settings  
n/a

Effect(s) tested  
n/a

Specify type of analysis:    ☐ Whole brain    ☐ ROI-based    ☒ Both

Anatomical location(s)    tumor volumes were manually assessed by the imaging core lab using RANO criteria

Statistic type for inference  
(See Eklund et al. 2016)  
n/a

Correction  
n/a

## Models & analysis

| n/a | Involved in the study |
|-----|----------------------|
| ☒ | ☐ Functional and/or effective connectivity |
| ☒ | ☐ Graph analysis |
| ☒ | ☐ Multivariate modeling or predictive analysis |

