## [Peer Review File · Nature]

Manuscript Title: A vaccine targeting mutant IDH1 in newly diagnosed glioma

Editorial Notes: *none*

Reviewer Comments & Author Rebuttals**Reviewer Reports on the Initial Version:**

Referee #1 (Remarks to the Author):

In the manuscript "A vaccine targeting mutant IDH1 in newly diagnosed glioma", Platten and colleagues conducted a multi-center phase I trial in 33 patients with grade 3 and 4 IDH mutant (R132H) astrocytomas using a IDH1R132H peptide vaccine. Majority of the patients (93%) mounted an immunological response, as measured by T cell and B cell response. Most notably, the immune response was independent of the MHC allele of the patient, overcoming a major limitation of peptide vaccination approach. In addition, contrast enhancing pseudoprogression was limited to patients eliciting an immune response and patients with progressive disease had neither an immune response nor pseudoprogression. ScRNA sequencing and TCR sequencing of a patient with pseudoprogression who underwent surgical resection of the lesion showed clonal expansion of mutant-specific CD4 T cells in the lesion, supporting a vaccine-induced T helper immune response. Proximity ligation assay of pre-treatment tumor tissue shows a high correlation between intratumoral IDH1R132H presentation on MHC II (before vaccination) and immune response following vaccination, indicating the presence of pre-existing mutant-specific CD4 T cell clones that selectively expand after the peptide vaccine.

The idea of using a vaccine targeting IDH1 mutation in humans is original and naturally follows the group's preclinical findings. The manuscript is significant as it presents a cogent picture of the immunological changes occurring after peptide vaccination and the resultant improvement in survival of the patients. The experimental approach is robust and the interpretation of data is valid and reliable. The quality of data compliments the importance of the study's findings. The manuscript is acceptable for publication, although the following minor changes could make it clearer to the reader.

The authors are encouraged to use language that better clarifies what subset of T cells are most likely responsible for the improvement in anti-tumor response according to the data. Although there has been a controversy surrounding the importance of a CD4 mediated anti-tumor response in the absence of a CD8 response, it is likely that cytotoxic or follicular helper-like CD4 T cells in the tumor are driving the anti-tumor response (with or without CD8 T cell involvement). The manuscript is ambiguous about the use of CD4 and/or CD8 cells in the immunological assays (for example ELISpot) and phrases such as "T cell response".

If only CD4 TCRs were reactive to mutant IDH1 (fig 4E), what were the top CD8 TCRs reactive to? Could CD8 TCR clonal expansion be secondary to an epitope-spread orchestrated by cytotoxic CD4 T cells? Or was it a result of cross-presentation of the vaccine?

Although the prior preclinical work by the group is cited, it would also be helpful to reiterate that CD8 T cells did not directly recognize the peptide vaccine in a complex with MHC I.

Other than these minor suggestions, the introduction, the conclusion and the results are expressed clearly.

Referee #2 (Remarks to the Author):

This is an important clinical immunotherapy study in patients with brain gliomas.

On the positive side is

- 1) the good choice of targeting by vaccination a mutation that appears to play a role in driving the malignant behavior. Hence its immune escape by loss or down-regulation of the mutant protein is less likely than with random mutations
- 2) this mutation is present in a high proportion of gliomas
- 3) the large majority of patients vaccinated shows a positive T cell immune response in PBMC. The mutation therefore shows strong immunogenicity and vaccination is therefore a therapeutic option for most patients with this mutation
- 4) There is circumstantial evidence of clinical efficacy
- 5) The T cell responses have been properly analysed by state of the art technology

The only obvious negative aspect is that this is not a randomized study.

However, the combined data can be considered sufficiently innovative and promising, showing an in depth and convincing account of an attractive and low-toxic novel approach to treat this therapy-resistant tumor with a dismal prognosis, as to be groundbreaking

Author Rebuttals to Initial Comments (please note that the authors have quoted the reviewers in black and responded in red):

Referee #1 (Remarks to the Author):

In the manuscript “A vaccine targeting mutant IDH1 in newly diagnosed glioma”, Platten and colleagues conducted a multi-center phase I trial in 33 patients with grade 3 and 4 IDH mutant (R132H) astrocytomas using a IDH1R132H peptide vaccine. Majority of the patients (93%) mounted an immunological response, as measured by T cell and B cell response. Most notably, the immune response was independent of the MHC allele of the patient, overcoming a major limitation of peptide vaccination approach. In addition, contrast enhancing pseudoprogression was limited to patients eliciting an immune response and patients with progressive disease had neither an immune response nor pseudoprogression. ScRNA sequencing and TCR sequencing of a patient with pseudoprogression who underwent surgical resection of the lesion showed clonal expansion of mutant-specific CD4 T cells in the lesion, supporting a vaccine-induced T helper immune response. Proximity ligation assay of pre-treatment tumor tissue shows a high correlation between intratumoral IDH1R132H presentation on MHC II (before vaccination) and immune response following vaccination, indicating the presence of pre-existing mutant-specific CD4 T cell clones that selectively expand after the peptide vaccine.

The idea of using a vaccine targeting IDH1 mutation in humans is original and naturally follows the group’s preclinical findings. The manuscript is significant as it presents a cogent picture of the immunological changes occurring after peptide vaccination and the resultant improvement in survival of the patients. The experimental approach is robust and the interpretation of data is valid and reliable.

The quality of data compliments the importance of the study's findings. The manuscript is acceptable for publication, although the following minor changes could make it clearer to the reader.

The authors are encouraged to use language that better clarifies what subset of T cells are most likely responsible for the improvement in anti-tumor response according to the data. Although there has been a controversy surrounding the importance of a CD4 mediated anti-tumor response in the absence of a CD8 response, it is likely that cytotoxic or follicular helper-like CD4 T cells in the tumor are driving the anti-tumor response (with or without CD8 T cell involvement). The manuscript is ambiguous about the use of CD4 and/or CD8 cells in the immunological assays (for example ELISpot) and phrases such as "T cell response".

Response 1: We thank the referee for this important comment. Indeed, we previously found that in an MHC-humanized mouse model, IDH1R132H-specific T cells were exclusively CD4-positive and that *ex vivo* recall T cell responses from patients with IDH1R132H-positive glioma were abrogated when blocking MHC class II (Schumacher, Nature, 2014). However, Pellegatta *et al.* (Pellegatta, Acta Neuropathologica Communications, 2015) reported the induction of IDH1R132H-specific T cells in C57Bl6 mice by using short (MHC class I-restricted) peptides. Therefore, overall, we cannot exclude that in some patients cytotoxic IDH1R132H-specific T cell responses might be generated.

In general, IDH1R132H-reactive B cell responses provide circumstantial evidence for CD4-restricted T helper cell responses (Foo, Nature Reviews Immunology, 2002). To provide further experimental evidence in this regard, we performed IDH1-vac-induced T cell sub-phenotyping by flow cytometry evaluating T cell effector cytokine production of peripheral IDH1R132H-reactive T cells from remaining patient samples with high MSS. We found predominant TNF- α , IFN- γ , and IL-17 cytokine production by T helper cells upon IDH1R132H *ex vivo* re-stimulation, indicative for Th1 and Th17 T helper cell subtypes (new Figure 2C). Neither IDH1R132H-specific IL-10 production by regulatory T cells nor TNF- α or IFN- γ production by cytotoxic T cells was observed (new Figure 2C).

If only CD4 TCRs were reactive to mutant IDH1 (fig 4E), what were the top CD8 TCRs reactive to? Could CD8 TCR clonal expansion be secondary to an epitope-spread orchestrated by cytotoxic CD4 T cells? Or was it a result of cross-presentation of the vaccine? Although the prior preclinical work by the group is cited, it would also be helpful to reiterate that CD8 T cells did not directly recognize the peptide vaccine in a complex with MHC I.

Response 2: We thank the referee for this interesting question. To address this question experimentally, we

- i) cloned two ID08 pseudoprogressive lesion-dominating CD8-positive TCRs ("TCR1": overall top abundant CD8-positive T cell clonotype; "TCR3": CD8-positive T cell clonotype with highest TNF- α /IFN- γ percentage (new Extended Data Figure 15), and
- ii) assessed *ex vivo* sorted CD8-positive endogenous T cells for IDH1R132H-reactivity (new Extended Data Figure 15).

In contrast to “TCR14”, none of the tested CD8-retrieved TCR-transgenic cells nor endogenous CD8-positive T cells showed IDH1R132H-reactivity (new Extended Data Figure 15 C,D). In this particular case, ID08-pseudoprogession was confirmed histologically by the (immune-)histological absence of tumor cells in the resected material. Therefore, in the absence of tumor cells, we were not able to identify the respective CD8-restricted tumor antigen that drives CD8-positive T cell activation and intratumoral expansion (new Extended Data Figure 15 A), retrospectively. However, the proinflammatory signature of the top CD8-clonotypes (new Extended Data Figure 15 A) is strongly suggestive for intralesional target recognition. Strong circumstantial evidence for this hypothesis is corroborated by mouse model systems applying CD4-restricted cancer vaccines that lead to the *de novo*-induction of CD8-positive T cell responses (Kreiter, Nature, 2015). This important concept is now included in the discussion section.

Other than these minor suggestions, the introduction, the conclusion and the results are expressed clearly.

Response 3: We thank the referee for this very positive feedback.

Referee #2 (Remarks to the Author):

This is an important clinical immunotherapy study in patients with brain gliomas. On the positive side is 1) the good choice of targeting by vaccination a mutation that appears to play a role in driving the malignant behavior. Hence its immune escape by loss or down-regulation of the mutant protein is less likely than with random mutations 2) this mutation is present in a high proportion of gliomas 3) the large majority of patients vaccinated shows a positive T cell immune response in PBMC. The mutation therefore shows strong immunogenicity and vaccination is therefore a therapeutic option for most patients with this mutation 4) There is circumstantial evidence of clinical efficacy 5) The T cell responses have been properly analysed by state of the art technology. The only obvious negative aspect is that this is not a randomized study. However, the combined data can be considered sufficiently innovative and promising, showing an in depth and convincing account of an attractive and low-toxic novel approach to treat this therapy-resistant tumor with a dismal prognosis, as to be groundbreaking

Response 4: We thank the referee for this very positive feedback. To further corroborate the low-toxicity of this novel approach, we included exploratory serum Luminex ELISA assays, demonstrating the absence of strong peripheral cytokine releases (Extended Data Figure 3), corroborating the concept of exclusive local inflammation at the injection site and in the CNS-tumor site.

Reviewer Reports on the First Revision:

Referee #1 (Remarks to the Author):

concerns have been addressed. Recommend acceptance.

Referee #2 (Remarks to the Author):

The authors have adequately responded to the issues raised. They have performed additional experiments with interesting outcomes regarding the question which type of CD4+ T cells has responded to the IDH1 mutation and whether in vivo epitope spreading to CD8+ T cells against unrelated antigens can have happened and they have added these informative data to the manuscript. They also added new data showing that no cytokine-release type toxicity was demonstrable from measuring serum cytokine levels

Referee #3 (Remarks to the Author):

The study conducted a single arm first-in-man phase I trial with 33 patients. The safety and efficacy summary results looks good. The statistical approaches they used for summary statistics is appropriate for the small sample size, such as Fisher's exact tests for categorical data analysis, and Kruskal-Wallis test for multiple group comparisons. For single cell RNA-seq data analysis, the visualization plot by UMAP looks good. Though I have a major concern for their Kaplan Meier curves:

Kaplan Meier curve is not appropriate to show two group comparison if the group defined by variable happened after day 0. It's not clear when is day 0 for "Time to death" and "Time to progression". I assume it's the day of receiving vaccine. For example, Figure 3B showed clearly separation between two groups defined by immune response Yes vs No. But immune response happened after receiving vaccine. That's a future information, so it's not appropriate to use this future information to classify patients status at day 0 (immune response Yes vs No), and they should use time-dependent Cox model instead (delete KM curves). Similarly, I assume MSS in Figure 3G is defined using information (duration of response ?) after receiving vaccine, thus it has similar issue.

Please find some references as below

- Therneau, T., Crowson, C., & Atkinson, E. (2013). Using time dependent covariates and time dependent coefficients in the cox model. *Red*, 2(1).
- Schultz, L. R., Peterson, E. L., & Breslau, N. (2002). Graphing survival curve estimates for time-dependent covariates. *International journal of methods in psychiatric research*, 11(2), 68-74.
- Bernasconi, D. P., Valsecchi, M. G., & Antolini, L. (2018). Non-parametric estimation of survival probabilities with a time-dependent exposure switch: application to (simulated) heart transplant data. *Epidemiology, Biostatistics and Public Health*, 15(3).

Author Rebuttals to First Revision (please note that the authors have quoted the reviewers in black and responded in red):

Referee #1 (Remarks to the Author):

concerns have been addressed. Recommend acceptance.

We thank the referee for her/his recommendation to accept our manuscript for publication in Nature.

Referee #2 (Remarks to the Author):

The authors have adequately responded to the issues raised. They have performed additional experiments with interesting outcomes regarding the question which type of CD4+ T cells has responded to the IDH1 mutation and whether in vivo epitope spreading to CD8+ T cells against unrelated antigens can have happened and they have added these informative data to the manuscript. They also added new data showing that no cytokine-release type toxicity was demonstrable from measuring serum cytokine levels

Thank you to the referee for the positive evaluation and for appreciation of additional data.

Referee #3 (Remarks to the Author):

The study conducted a single arm first-in-man phase I trial with 33 patients. The safety and efficacy summary results looks good. The statistical approaches they used for summary statistics is appropriate for the small sample size, such as Fisher's exact tests for categorical data analysis, and Kruskal-Wallis test for multiple group comparisons. For single cell RNA-seq data analysis, the visualization plot by UMAP looks good. Though I have a major concern for their Kaplan Meier curves:

Kaplan Meier curve is not appropriate to show two group comparison if the group defined by variable happened after day 0. It's not clear when is day 0 for "Time to death" and "Time to progression". I assume it's the day of receiving vaccine. For example, Figure 3B showed clearly separation between two groups defined by immune response Yes vs No. But immune response happened after receiving vaccine. That's a future information, so it's not appropriate to use this future information to classify patients status at day 0 (immune response Yes vs No), and they should use time-dependent Cox model instead (delete KM curves). Similarly, I assume MSS in Figure 3G is defined using information (duration of response ?) after receiving vaccine, thus it has similar issue.

Please find some references as below

- Therneau, T., Crowson, C., & Atkinson, E. (2013). Using time dependent covariates and time dependent coefficients in the cox model. *Red*, 2(1).
- Schultz, L. R., Peterson, E. L., & Breslau, N. (2002). Graphing survival curve estimates for time-dependent covariates. *International journal of methods in psychiatric research*, 11(2), 68-74.
- Bernasconi, D. P., Valsecchi, M. G., & Antolini, L. (2018). Non-parametric estimation of survival probabilities with a time-dependent exposure switch: application to (simulated) heart transplant data. *Epidemiology, Biostatistics and Public Health*, 15(3).

We thank the referee for her/his positive remarks, for detailed review of statistics, and suggestions for survival data visualization using time-dependent covariates.

Indeed, we agree with the referee that for Kaplan Meier curves in Figures 3b (immune response yes vs. no) and 3g (MSS below vs. above median), these group-defining variables happened after day 0. Day 0 is the time point of first diagnosis; we have made that clear for all Kaplan Meier curves in Figure 3b and g and ED Figure 9, now ED Figure 6, by changing the

axis labels to “Time to Progression/Death since first diagnosis (months) (x-axes) and “Probability” (y-axes) in the revised version of the manuscript. We have substantially revised Figures 3b and 3g. Survival data according to immune response and MSS score are now visualized as Simon and Makuch Plots of overall and progression-free survival probabilities according to the time-dependent covariate IDH1-vac-induced IR, or mutation-specificity score, respectively. We have adapted the respective figure legends accordingly and made that clear in the figure.